# Limited Preference Data? Learning Better Reward Model with Latent Space Synthesis

**Leitian Tao[1]    Xuefeng Du[2]    Sharon Li[1]**

[1]Department of Computer Sciences, University of Wisconsin-Madison
[2]College of Computing and Data Science, Nanyang Technological University
leitiantao@cs.wisc.edu, xuefeng.du@ntu.edu.sg, sharonli@cs.wisc.edu

## Abstract

Reward modeling, crucial for aligning large language models (LLMs) with human preferences, is often bottlenecked by the high cost of preference data. Existing textual data synthesis methods are computationally expensive. We propose a novel framework **LENS** for synthesizing preference data directly in the LLM's latent embedding space. Our method employs a Variational Autoencoder (VAE) to learn a structured latent representation of response embeddings. By performing controlled perturbations in this latent space and decoding back to the embedding space, we efficiently generate diverse, semantically consistent synthetic preference pairs, bypassing costly text generation and annotation. We provide theoretical guarantees that our synthesized pairs approximately preserve original preference ordering and improve reward model generalization. Empirically, our latent-space synthesis significantly outperforms text-based augmentation on standard benchmarks, achieving superior results while being 18× faster in generation and using a 16,000× smaller model. Our work offers a scalable and effective alternative for enhancing reward modeling through efficient data augmentation. Code is publicly available at https://github.com/deeplearning-wisc/lens.

## 1 Introduction

As large language models (LLMs) are increasingly deployed in settings that interact with or influence people [1, 2, 3, 4], translating human feedback into a reward function has become a cornerstone of AI development [5]. Reward modeling, which learns to assign higher scores to preferred responses over rejected ones, is a critical component in post-training and decision-making systems [6, 7, 8, 9]. A strong reward model can be used not only to supervise language model behavior but also to support high-leverage tasks like rejection sampling, preference ranking, and quality estimation.

Despite its importance, reward modeling remains bottlenecked by data collection and computational cost. Human preference labels are expensive and time-consuming to collect at scale. To address the high cost of collecting human preference data, researchers have explored textual space synthesis approaches [10, 11, 12, 13, 14]. As shown in Figure 1, these methods typically involve a two-stage process: first, using LLMs to generate multiple diverse responses to the same prompt; then, employing an auxiliary LLM to annotate these responses by creating pairwise preference data, which is subsequently used for reward model training. However, this approach faces significant computational challenges. The response generation phase requires substantial computational resources to produce diverse, high-quality candidates. The preference annotation phase is also resource-intensive, as it requires running LLMs to evaluate and rank each response pair, which scales quadratically with the

39th Conference on Neural Information Processing Systems (NeurIPS 2025).

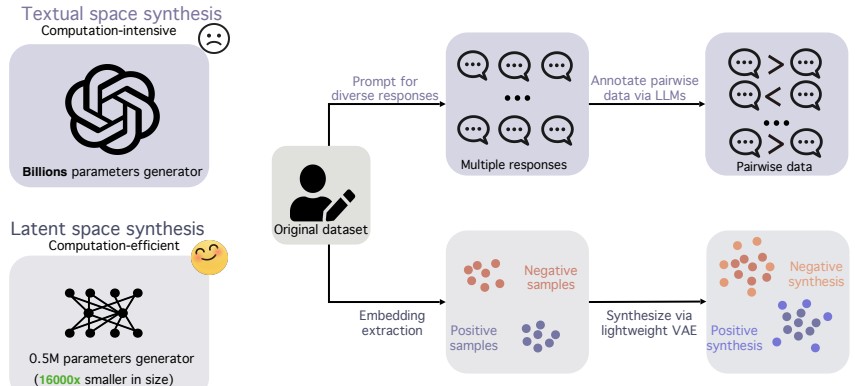

Figure 1: Comparison of textual space synthesis (top) and latent space synthesis (bottom). Latent space synthesis operates on embeddings, offering significant computational advantages. Best viewed in color.

number of responses per prompt. These challenges raise the question: ***Given limited preference data, can we efficiently expand the dataset to improve reward modeling***?

To address these challenges, we propose a new framework–**LENS** (Latent EmbeddiNg for Synthesis)—which synthesizes preference data *in the latent space* rather than the textual space. LENS bypasses the computational expense of text generation, avoids complex prompt engineering, and leverages the semantic structure already captured by the language model embeddings. To model and generate plausible preference data, we leverage a Variational Autoencoder (VAE), a generative model that learns a smooth and structured distribution in its latent space. The VAE is particularly well-suited for this task: mapping each input to a distribution over latent variables rather than a fixed point enables localized sampling that preserves semantic consistency while introducing meaningful diversity—which is critical for generating synthetic preference pairs. In particular, we synthesize new preference pairs by performing controlled perturbations in the learned latent space of VAE, which can then be decoded back to the LLM embedding space. This creates an augmented dataset to enhance reward modeling without requiring additional human annotation and bypassing expensive text generation.

Importantly, our approach comes with theoretical guarantees. We show that synthetic preference pairs generated by LENS preserve the original preference ordering, up to a bounded error that depends on the noise level and reconstruction quality of VAE (Theorem 1). This establishes that preference relationships are maintained after latent-space perturbation and decoding. Moreover, we prove that augmenting the original training set with these synthetic samples reduces the error upper bound in reward modeling by effectively increasing the sample size while maintaining preference consistency (Theorem 2). Beyond theoretical understanding, we further validate our method on widely used reward modeling benchmarks, demonstrating that latent-space synthesis consistently outperforms text-based synthesis approaches. Notably, our method requires significantly fewer computational resources to sample new preference data, requiring only 0.5M parameters compared to billions of LLM parameters for the text-based synthesis method, reducing the generator size by $16{,}000\times$. These efficiency gains make our approach highly scalable and practical for real-world deployment, especially in resource-constrained settings. We summarize our contributions below:

1. We propose a novel framework LENS for synthesizing preference data directly in the latent embedding space of language models, enabling efficient reward modeling without the need for text generation or heavy prompt engineering.

2. We design a contrastive VAE that learns to generate diverse yet semantically meaningful synthetic embeddings and provide a theoretical analysis showing how latent-space synthesis improves the generalization of reward modeling.

3. We demonstrate strong empirical results across reward modeling benchmarks, achieving superior performance to text-based augmentation while being 18× faster and 16,000× smaller in model size, with detailed ablations validating design choices.

## 2 Preliminaries

Large language models map a natural language prompt $x \in \mathcal{X}$ to a generated response $y \in \mathcal{Y}$, defining a probability distribution $p_\theta(y \mid x)$ over the vocabulary space $\mathcal{V}$. The reward model acts as an auxiliary function that evaluates the quality of responses, which is trained to produce a higher

score for the preferred response given a query. Reward modeling relies on preference-labeled data, typically collected as pairwise comparisons. Formally, we define the preference data below.

**Definition 1** (**Preference Data.**). *Consider two responses* $y^+, y^-$ *for an input prompt* $x$, *we denote* $y^+ \succ y^-$ *if* $y^+$ *is preferred over* $y^-$. *We call* $y^+$ *the preferred response and* $y^-$ *the rejected response. Each triplet* $(x, y^+, y^-)$ *is referred to as a preference. Furthermore, the empirical dataset* $\mathcal{D} = \{(x_i, y_i^+, y_i^-)\}_{i=1}^N$ *consists of* $N$ *such triplets sampled from a preference distribution* $\mathcal{P}$.

**Reward modeling objective.** Reward modeling learns a function mapping, which takes in the prompt $x$ and response $y$ and outputs a scalar value $r(x, y)$ signifying the reward. A preferred response should receive a higher reward, and vice versa. Based on the Bradley–Terry model [15], the reward function is optimized over a dataset of human preferences, with the following objective:

$$\mathcal{L}_{RM}^{\mathcal{D}} = -\mathbb{E}_{(x,y^+,y^-)\in\mathcal{D}}[\log \sigma(r(x, y^+) - r(x, y^-))], \tag{1}$$

where $\sigma$ denotes the sigmoid function. This objective encourages the reward function to produce the observed orderings. Once trained, the reward model can be used to score future responses. *Training high-quality reward models remains heavily bottlenecked by the cost of collecting large-scale human preference data, which is both time-consuming and labor-intensive*. Our work hence focuses on the practical setting where $N$ is moderately small. Recent work has explored augmenting preference datasets through textual response synthesis and LLM-based annotation [12, 13, 14], but such pipelines are computationally intensive, often requiring powerful models for both generation and labeling. These challenges motivate our approach to synthesize preference data directly in the latent embedding space of the language model, bypassing the need for costly generation and annotation and thus offering stronger efficiency and scalability.

## 3   Methodology

In this section, we introduce our methodology to efficiently generate preference data by synthesizing new samples directly in the latent space, rather than in the text space. This paradigm shift avoids expensive text generation, mitigates complex prompt engineering, and leverages the semantic structure already captured by the language model. Our framework consists of three main stages: (1) training a variational autoencoder with divergence learning on response embeddings (see Section 3.1), (2) generating synthetic preference pairs in the learned latent embedding space (see Section 3.2), and (3) training a reward model on a combination of original and synthesized preference data (see Section 3.3). The subsections below describe each component in detail.

### 3.1   Variational Autoencoder with Divergence Learning

We begin by extracting embeddings for the language model's responses. Given a preference dataset $\mathcal{D} = \{(x_i, y_i^+, y_i^-)\}_{i=1}^N$ as defined in Section 2, where $x_i$ is the prompt, $y_i^+$ is the preferred response, and $y_i^-$ is the non-preferred response, we extract the LLM embedding vectors for each preference triplet in the dataset:

$$\mathbf{e}_i^{\pm} = \text{LLM}_{\text{embed}}(x_i, y_i^{\pm}), \tag{2}$$

where $\text{LLM}_{\text{embed}}$ represents the embedding extraction function that processes each prompt-response pair through the language model and returns the final hidden state representation at the last layer. These embeddings capture the semantic representation of the responses and serve as the starting point for our VAE-based synthesis.

**Learning latent representations with divergence-aware VAE.** To enhance the reward model, we synthesize additional training pairs by leveraging a VAE [16], a generative model that learns a probabilistic latent representation of preference data. VAE is a natural choice due to its ability to learn a smooth and structured latent space, enabling the sampling of diverse yet semantically coherent synthetic preference data. In particular, VAEs map each input to a distribution over latent variables rather than a single point. This is crucial in our context, as it enables diverse yet semantically consistent sampling around a given embedding, which is important for generating diverse synthetic preference data for better reward modeling.

Specifically, given a dataset of preference embeddings $\mathcal{E} = \{\mathbf{e}_i^+, \mathbf{e}_i^-\}_{i=1}^N$, where each $\mathbf{e}_i^+$ and $\mathbf{e}_i^-$ denotes the original LLM embedding of a preferred and non-preferred response, the VAE encoder

$\phi : \mathbb{R}^d \rightarrow \mathbb{R}^{2d_{\text{VAE}}}$ transforms each $d$-dimensional LLM embedding into the posterior Gaussian parameter vectors $\boldsymbol{\mu}_\phi | \mathbf{e} \in \mathbb{R}^{d_{\text{VAE}}}$ and $\boldsymbol{\sigma}_\phi^2 | \mathbf{e} \in \mathbb{R}^{d_{\text{VAE}}}$, and $d_{\text{VAE}}$ is the dimension of the VAE latent space. The posterior distributions for the positive and negative embeddings are parameterized as:

$$q_\phi(\mathbf{z}|\mathbf{e}^+) = \mathcal{N}(\mathbf{z}; \boldsymbol{\mu}_\phi(\mathbf{e}^+), \boldsymbol{\sigma}_\phi(\mathbf{e}^+)^2 \cdot \mathbf{I}), \quad q_\phi(\mathbf{z}|\mathbf{e}^-) = \mathcal{N}(\mathbf{z}; \boldsymbol{\mu}_\phi(\mathbf{e}^-), \boldsymbol{\sigma}_\phi(\mathbf{e}^-)^2 \cdot \mathbf{I}). \quad (3)$$

The decoder then reconstructs the embeddings from samples in the latent space. This reconstruction process, denoted as $\hat{\mathbf{e}} = g_\theta(\mathbf{z})$, transforms the latent representation back into the original embedding space. The VAE loss for each sample (either a positive or a negative embedding) is given by:

$$\mathcal{L}_{\text{VAE}}(\mathbf{e}) = \mathcal{L}_{\text{recon}}(\mathbf{e}, \hat{\mathbf{e}}) + \beta \cdot D_{\text{KL}}\left(q_\phi(\mathbf{z}|\mathbf{e}) \,\|\, p(\mathbf{z})\right), \quad (4)$$

Where $\mathcal{L}_{\text{recon}}$ denotes a reconstruction loss between original and reconstructed embeddings, and $p(\mathbf{z}) = \mathcal{N}(\mathbf{0}, \mathbf{I})$ is the standard isotropic Gaussian prior. The hyperparameter $\beta$ balances reconstruction quality and latent regularization.

To encourage the model to learn discriminative representations, we introduce a divergence term to maximize the separation between the latent distributions of positive and negative samples:

$$\mathcal{L}_{\text{divergence}} = -\frac{1}{N} \sum_{i=1}^{N} W_2\left(q_\phi(\mathbf{z}^+|\mathbf{e}_i^+), \, q_\phi(\mathbf{z}^-|\mathbf{e}_i^-)\right), \quad (5)$$

Where $W_2$ denotes the Wasserstein distance. The overall objective is defined as:

$$\mathcal{L}_{\text{total}} = \frac{1}{N} \sum_{i=1}^{N} \left[\mathcal{L}_{\text{VAE}}(\mathbf{e}_i^+) + \mathcal{L}_{\text{VAE}}(\mathbf{e}_i^-)\right] + \gamma \cdot \mathcal{L}_{\text{divergence}}, \quad (6)$$

with hyperparameter $\gamma$ controlling the importance of divergence term. We provide ablations on the impact of both $\beta$ and $\gamma$ in Section 5.3 and Appendix A.2.

## 3.2 Latent Space Synthesis

**Latent space sampling.** Once the VAE has been trained on preference embeddings, we synthesize new data by performing controlled perturbations on the known pairwise responses in the learned latent space. The goal is to generate new embeddings that maintain the semantic representation of the original responses while preserving preference relationships. For each embedding, multiple noisy variants are generated by adding Gaussian noise to its latent vector $\mathbf{z}_i^{\pm}$; these noisy latent vectors are subsequently decoded to generate new embeddings $\hat{\mathbf{e}}_{i,j}^{\pm}$:

$$\hat{\mathbf{e}}_{i,j}^{\pm} = g_\theta(\mathbf{z}_i^{\pm} + \boldsymbol{\eta}_{i,j}^{\pm}), \quad \text{where } \boldsymbol{\eta}_{i,j}^{\pm} \sim \mathcal{N}(0, \sigma_{\text{noise}}^2 \mathbf{I}), \quad (7)$$

where $j$ indexes the number of synthetic samples per original embedding, and $\sigma_{\text{noise}}$ is a hyperparameter controlling the noise level. To ensure quality and consistency, we select the top-$k$ synthetic latent codes $(\mathbf{z}_i^{\pm} + \boldsymbol{\eta}_{i,j}^{\pm})$ based on their likelihood under the learned distributions, before decoding them.

**Forming synthetic preference pairs.** To expand the training data, we form synthetic preference pairs by pairing each synthesized or original preferred embedding with both synthesized and original non-preferred embeddings, and vice versa. This compositional pairing strategy gets an augmented dataset:

$$\mathcal{E}_{\text{aug}} = \left\{(\tilde{\mathbf{e}}^+, \tilde{\mathbf{e}}^-) \,\middle|\, \tilde{\mathbf{e}}^+ \in \mathcal{E}^+ \cup \mathcal{E}_{\text{synth}}^+, \, \tilde{\mathbf{e}}^- \in \mathcal{E}^- \cup \mathcal{E}_{\text{synth}}^-\right\}, \quad (8)$$

where $\mathcal{E}_{\text{synth}}^+ = \{\hat{\mathbf{e}}_{i,j}^+\}$ and $\mathcal{E}_{\text{synth}}^- = \{\hat{\mathbf{e}}_{i,j}^-\}$ denote the sets of synthesized positive and negative embeddings, respectively.

## 3.3 Reward Modeling with the Synthesized Preference Data

The augmented dataset is used to train a reward model that can distinguish preferred data from rejected ones. During optimization, these pairs contribute to the reward modeling objective:

$$\mathcal{L}_{RM}^{\mathcal{E}_{\text{aug}}} = -\mathbb{E}_{(\tilde{\mathbf{e}}^+, \tilde{\mathbf{e}}^-) \in \mathcal{E}_{\text{aug}}}\left[\log \sigma(r_o(\tilde{\mathbf{e}}^+) - r_o(\tilde{\mathbf{e}}^-))\right], \quad (9)$$

where the function $r_o$ is a lightweight MLP mapping the embedding to the reward score. This loss encourages the model to assign higher scores to preferred embeddings and vice versa. We adopt an

MLP architecture for reward modeling because recent advances in reward modeling have highlighted such training approach as a lightweight yet effective alternative to full fine-tuning [17, 18, 19, 20, 21, 22, 23]. In Section 5, we show that this embedding-based reward modeling approach outperforms full fine-tuning methods trained on text-space augmentations, while requiring significantly less computing. By training on both real and synthesized pairs, the reward model benefits from exposure to a wider range of preference pairs, ultimately leading to better generalization.

## 4  Theoretical Analysis

In this section, we provide a theoretical analysis to support our proposed algorithm. As an overview, Theorem 1 analyzes the *quality* of the synthetic preference pairs under the best possible reward function. We then provably investigate the *learnability* of the reward model trained with the synthesized preference data in Theorem 2, demonstrating that it can be better than the reward model trained without synthesis under certain regulatory conditions. We specify several mild assumptions and necessary notations for our theorems in Appendix B.1. Due to space limitations, we omit unimportant constants and simplify the statements of our theorems. We defer the full formal statements in Appendix B.2. All proofs can be found in Appendix B.3.

### 4.1  Analysis on Synthesis Quality

We first analyze the quality of the synthetic preference embeddings $(\hat{\mathbf{e}}^+, \hat{\mathbf{e}}^-)$ by the best possible reward MLP function that serves as an ideal evaluator trained over the original preference data distribution. Specifically, let $\mathcal{R}_o$ denote the hypothesis space of the reward MLP model, and $\mathcal{P}_e$ as the embedding distribution of the original preference dataset, the best possible reward function is formulated as $r_o^* = \operatorname{argmin}_{r_o \sim \mathcal{R}_o} \mathbb{E}_{(\mathbf{e}^+, \mathbf{e}^-) \sim \mathcal{P}_e} [-\log \sigma(r_o(\mathbf{e}^+) - r_o(\mathbf{e}^-))]$. Based on $r_o^*$, the reward difference between the synthesized positive and negative pairs $r_o^*(\hat{\mathbf{e}}^+) - r_o^*(\hat{\mathbf{e}}^-)$ has the following bound:

> **Theorem 1.** *(Informal). Under mild conditions, for any preference LLM embedding $\mathbf{e} \sim \mathcal{E}$, sample a latent vector $\mathbf{z} \sim q_\phi(\cdot|\mathbf{e})$, if there exists a constant $\epsilon_{\mathrm{rec}}$ that satisfies $\|g_\theta(q_\phi(\mathbf{z}|\mathbf{e})) - \mathbf{e}\| \le \epsilon_{\mathrm{rec}}$, then with a high probability, for any synthesized preference embedding pairs $(\hat{\mathbf{e}}^+, \hat{\mathbf{e}}^-)$, their reward difference when evaluated by the best possible reward MLP function $r_o^*$ is lower bounded by*
>
> $$r_o^*(\hat{\mathbf{e}}^+) - r_o^*(\hat{\mathbf{e}}^-) \ge r_o^*(\mathbf{e}^+) - r_o^*(\mathbf{e}^-) - \mathcal{O}(\sigma_{\mathrm{noise}}\sqrt{d_{\mathrm{VAE}}}) - \mathcal{O}(\epsilon_{\mathrm{rec}}), \qquad (10)$$
>
> *where $d_{\mathrm{VAE}}$ is the dimension of the VAE latent space, and $(\mathbf{e}^+, \mathbf{e}^-)$ is the corresponding preference embedding pair from which $(\hat{\mathbf{e}}^+, \hat{\mathbf{e}}^-)$ is synthesized.*

**Implications.** The theorem states that under mild assumptions, the rewards of synthesized positive samples have a bounded margin compared to those of synthetic negative ones. If the following conditions hold: 1) the VAE is well trained so that the reconstruction error is small; 2) the injected noise magnitude $\sigma_{\mathrm{noise}}$ and the VAE latent dimension $d_{\mathrm{VAE}}$ is small (e.g., we use 16 in practice); 3) the reward margin on the original preference embedding pairs $r_o^*(\mathbf{e}^+) - r_o^*(\mathbf{e}^-)$ is large and bigger than 0, then the lower bound will be tight. **We verify these conditions hold empirically in Appendix B.5**.

### 4.2  Analysis on Reward Model Learnability

In this section, we provide the learnability analysis for the reward MLP model that is trained with the augmented dataset $\mathcal{E}_{\mathrm{aug}}$ (Section 3.3). Our results below show that the reward model trained with $\mathcal{E}_{\mathrm{aug}}$ can achieve a smaller estimation error compared to the model trained with the original preference dataset $\mathcal{E}$ under certain regulatory conditions.

Formally, the estimation error of a reward MLP function trained on preference dataset $\mathcal{E}$ is defined as $\zeta_\mathcal{E} = \mathcal{L}_{RM}^{\mathcal{P}_e}(\hat{r}_\mathcal{E}) - \mathcal{L}_{RM}^{\mathcal{P}_e}(r_o^*)$ where $\mathcal{L}_{RM}^{\mathcal{P}_e}$ is defined over distribution $\mathcal{P}_e$ and and can be calculated by Equation 9. The empirical risk minimizer $\hat{r}_\mathcal{E}$ on the original preference dataset is formulated as $\hat{r}_\mathcal{E} = \operatorname{argmin}_{r_o \in \mathcal{R}_o} \mathcal{L}_{RM}^{\mathcal{E}}(r_o)$. Then, we have

**Theorem 2.** *(Informal). Let $kN$ be the size of the augmented preference dataset. If the reconstruction error in Theorem 1 decays as $\epsilon_{\mathrm{rec}} = \mathcal{O}(N^{-p})$ where $p > 0$, and with probability at least $1 - \delta$ and a universal constant $C > 0$, if we further require $N > \left(C\sqrt{d + \log(1/\delta)}\right)^{\frac{1}{1/2-p}}$, then there exists a constant $k_0 > 1$ such that for all $k \geq k_0$, the following estimation error condition of the reward model hold:*

$$\zeta_{\mathcal{E}_{aug}} < \zeta_{\mathcal{E}}. \tag{11}$$

In Theorem 2, we reflect the necessary condition for the latent space synthesis to help reward modeling, and it requires the sample size of the original preference dataset to be larger than a constant. Appendix B.5 empirically demonstrates that we only need a small number of original preference pairs in order to guarantee a better performance, which is easy to satisfy in practice.

## 5 Experiments

In this section, we evaluate our latent-based synthesis approach for reward modeling, comparing LENS against baseline approaches across different scales and sample sizes, followed by ablation studies to analyze the impact of various components of our methodology.

### 5.1 Experimental Settings

**Model and datasets.** We use the Llama-3.1-8B-Instruct [24] as the base model. For experiments, we use two preference datasets: (1) HH-RLHF [25], which contains human preference pairs focused on helpfulness and harmlessness; and (2) the TL;DR summarization [26], consisting of preference pairs for Reddit post summarization. To explore the effectiveness of using synthesis to extend the training dataset in a sample-limited scenario, we subsample 1,000 samples as seed samples. In our ablations, we extensively verify different LLM backbones and different numbers of seed samples. Full experimental configurations are included in Appendix A.1.

**Evaluation.** We evaluated the effectiveness of different reward models using Best-of-N (BoN) sampling following previous work [23, 27, 28]. For each prompt in the test set, we generate $n$=16 candidate responses from the base model. The trained reward model then ranks these candidates, and we select the highest-scoring response. We report the average reward score of these selected responses as evaluated by a held-out gold reward model trained on diverse and high-quality datasets. In our experiments, the Skywork [29] model serves as the gold reward as the ground truth quality.

**Baselines.** The traditional method of collecting pairwise preference data for reward modeling focuses on **text space synthesis**, where different responses to the same prompt are gathered and then labeled as preferred or non-preferred by human annotators or large language models. To establish baselines, we first use the base model to generate multiple responses for each prompt in the training set. For reproducibility, we employ a well-trained reward model to score these responses and determine preference rankings. We compare our latent space synthesis approach against several text-based methods, including (1) fully fine-tuned models that update all parameters with text-space preference data; (2) Low-rank adaptation techniques that modify only a subset of parameters; and (3) Embedding-based approaches that keep the backbone fixed while training an MLP reward head. Some works propose using the model itself to label diverse samples (which avoids introducing an additional reward model). The corresponding baselines include: (4) Self-rewarding [12]: the model as a judge to rank pairwise responses. (5) Self-evolved [13]: using the reward to ranking the pairwise responses. (6) IPO [13]: take the likelihood of the different responses to decide the preference. Finally, for **latent-space synthesis baselines**, we consider (7) direct noise addition to LLM embeddings and (8) Gaussian sampling that assumes a normal distribution of LLM embeddings without the structured learning of our VAE approach. These baselines provide comprehensive comparison points across both textual and latent synthesis paradigms, allowing us to evaluate the proposed method.

### 5.2 Main Results

**Synthesis in latent space significantly boosts reward model performance.** Table 1 presents the main results, demonstrating that our VAE-based latent space synthesis approach consistently outperforms textual space synthesis methods across both the HH-RLHF and TL;DR datasets at various

Table 1: **Main results**. Our latent-space synthesis approach outperforms text-based synthesis on both HH-RLHF and TL;DR benchmarks. We report the mean and variance of our results with *three different runs*. $2\times$, $4\times$, and $8\times$ denote the augmentation scale.

| Method | HH-RLHF | | | | TL;DR | | | |
|---|---|---|---|---|---|---|---|---|
| | Original | $2\times$ | $4\times$ | $8\times$ | Original | $2\times$ | $4\times$ | $8\times$ |
| **Textual Synthesis** | | | | | | | | |
| Fully fine-tune | 1.49 | 1.57 | 1.78 | 1.93 | 0.69 | 0.84 | 0.97 | 1.23 |
| Low rank adaptation | 1.28 | 1.48 | 1.52 | 1.61 | 0.57 | 0.87 | 0.92 | 1.15 |
| Embedding MLP | 1.43 | 1.51 | 1.62 | 1.73 | 0.78 | 0.94 | 1.02 | 1.11 |
| Self-rewarding [12] | 1.49 | 1.48 | 1.59 | 1.77 | 0.69 | 0.78 | 0.92 | 0.95 |
| Self-evolved [13] | 1.49 | 1.42 | 1.54 | 1.63 | 0.69 | 0.72 | 0.79 | 0.75 |
| IPO [14] | 1.49 | 1.37 | 1.25 | 1.32 | 0.69 | 0.63 | 0.67 | 0.62 |
| **Latent Space Synthesis** | | | | | | | | |
| Direct perturbation | 1.43 | 1.46 | 1.32 | 1.46 | 0.78 | 0.81 | 0.84 | 0.79 |
| Gaussian sampling | 1.43 | 1.23 | 1.12 | 0.94 | 0.78 | 0.64 | 0.53 | 0.43 |
| LENS (Ours) | 1.43 | $\mathbf{1.86}^{\pm 0.04}$ | $\mathbf{1.94}^{\pm 0.06}$ | $\mathbf{2.20}^{\pm 0.12}$ | 0.78 | $\mathbf{1.25}^{\pm 0.03}$ | $\mathbf{1.44}^{\pm 0.05}$ | $\mathbf{1.48}^{\pm 0.07}$ |

data augmentation scales. We highlight several key observations. *First*, even without augmentation, the embedding-based MLP reward model performs competitively, achieving scores close to the fully fine-tuned model on two datasets. This indicates that LLM embeddings inherently capture significant preference information, validating the foundation of embedding-based reward modeling. *Second*, when applying augmentation, our VAE-based method shows substantial gains. For instance, at $4\times$ augmentation on HH-RLHF, our method achieves a reward score of 1.96, significantly outperforming the strongest textual synthesis baseline (Fully fine-tuned at 1.78). Similarly, on TL;DR at $4\times$ augmentation, our method reaches 1.42, compared to 0.97 for the fully fine-tuned textual approach. The advantage becomes even more pronounced at $8\times$ augmentation (2.17 *vs.* 1.93 on HH-RLHF; 1.46 *vs.* 1.23 on TL;DR). These results underscore the effectiveness of generating synthetic data within a learned latent space. *Lastly*, we compare our approach to simpler latent space baselines. Direct perturbation yields inconsistent results, and Gaussian-based sampling leads to significant performance degradation. This comparison highlights the critical role of the VAE in learning a structured latent representation that allows for the generation of diverse yet semantically meaningful and preference-preserving synthetic data, which naive LLM latent space manipulations fail.

**Latent space synthesis significantly reduces computational costs.** We compare the computational cost between textual and latent space synthesis approaches for the $8\times$ augmentation on HH-RLHF. As shown in Table 2, our latent space approach yields substantial savings. It requires only 0.5M parameters compared to 8B for the text-based method—a 16,000× reduction.

| Metric | Textual | Latent | Reduction |
|---|---|---|---|
| Generation time $\downarrow$ | 3.6h | **0.2h** | 18× |
| Model size $\downarrow$ | 8B | **0.5M** | 16,000× |
| Total runtime $\downarrow$ | 5.2h | **0.4h** | 13× |

Table 2: Computational efficiency comparison

This translates to a 13× speedup in total processing time (from 5.2 hours to 0.4 hours on a single A100 GPU). Sample synthesis time specifically decreases from 3.6 hours to 0.2 hours ($\downarrow$18× reduction). These efficiency gains make our approach highly scalable and practical for real-world deployment, especially in resource-constrained environments. Moreover, the lightweight nature of our method allows preference synthesis to be performed rapidly, especially in scenarios where collecting human preference data can be slow and expensive.

**LENS generalizes well across different model families and tasks.** Our latent space synthesis method demonstrates strong generalization capabilities across diverse model architectures and datasets. As shown in Table 4 in the Appendix, when applied to various base models from different families and scales including Gemma-2B [30], Llama-3.2-3B [24], Mistral-7B [31], Qwen-2.5-7B [32], and Llama-3.1-8B [24], our approach consistently yields higher reward scores compared to both the original baseline and textual space synthesis at a $4\times$ augmentation rate. For instance, with Llama-3-3B, latent synthesis achieves a reward of 0.73, significantly higher than textual synthesis (0.58) and the original baseline (0.44). Furthermore, the main results in Table 1 confirm this generalizability across different tasks. Our VAE-based method significantly outperforms textual synthesis methods on both the HH-RLHF and TL;DR datasets, particularly at higher augmentation factors. This consistent performance across different models and datasets underscores the broad applicability and effectiveness of synthesizing preference data in the learned latent space.

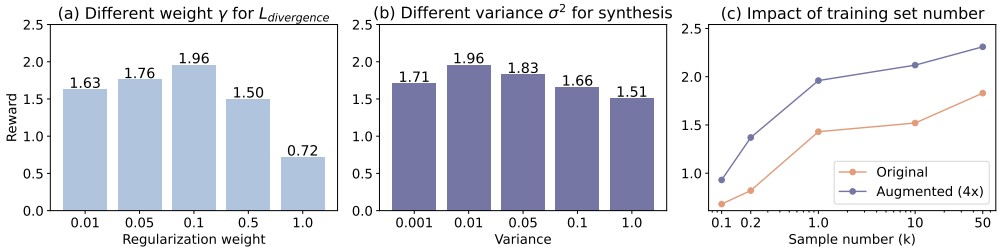

Figure 2: (a) Effect of weight of $\mathcal{L}_{\text{divergence}}$. (b) Effect of noise variance $\sigma^2$ during synthesis. (c) Ablation on the number of initial training samples (in thousands).

**Our reward model improves SFT via rejection sampling.** We further evaluate how reward models trained with different synthesis methods impact downstream supervised fine-tuning (SFT). We use a held-out set of 1,000 HH-RLHF prompts (different samples that are not used in reward model training). For each prompt, we generate 16 candidate completions using Llama-3.1-8B-Instruct. Following [27], the trained reward model performs rejection sampling, selecting the highest-scoring response as the

| Synthesis | Reward | GPT-4 eval |
|---|---|---|
| Textual | 1.62 | 39% |
| Latent | **1.96** | **61%** |

Table 3: Comparison of SFT model performance using rejection sampling under two different reward models.

target for SFT. We compare SFT models trained using targets selected by two reward models: one trained with text-based synthetic data, and the other with our embedding-based latent synthesis. Both reward models use $4\times$ augmentation, and the SFT models share identical training configurations (Appendix A.2) apart from the selected targets. As shown in Table 3, using the reward model trained with latent synthesis leads to higher SFT performance and a better win rate in GPT-4 as a judge [33] pairwise evaluation (61% vs. 39%). These results demonstrate that our latent synthesis not only accelerates reward modeling but also enables better SFT outcomes by guiding sample selection more effectively.

## 5.3 Ablation Studies

**Effect of divergence loss weight $\gamma$.** We ablate the impact of the contrastive loss weight $\gamma$, which controls the degree of separation between the VAE latent distributions of preferred and non-preferred data. As shown in Figure 2a, increasing $\gamma$ from 0 to 0.1 improves the quality of the learned latent space, leading to stronger separation and higher downstream reward performance. However, excessively large $\gamma$ values (e.g., 0.5 or 1.0) result in over-separation, ultimately degrading performance. We hypothesize that when the divergence loss is too strong, the latent embeddings of positive and negative responses become trivially distinguishable. As a result, the synthetic preference pairs constructed from such over-separated distributions are too easy for the reward model, diminishing the benefit.

The visualization in Figure 3 supports this, showing that moderate regularization (e.g., $\gamma = 0.1$) achieves a balanced latent geometry, where preferred (orange) and non-preferred (purple) responses are well-separated yet still diverse. This underscores the importance of setting $\gamma$ mildly to avoid degenerate solutions and preserve the richness of preference training data.

**Effect of synthesis noise $\sigma^2$.** Figure 2b demonstrates how the variance of noise added during latent space synthesis affects model performance. Moderate noise levels ($\sigma^2 = 0.01$) yield optimal results with a reward score of 1.96. This aligns with our theoretical insight that the quality of synthetic preferences depends on the noise term. When the noise is too small ($\sigma^2 = 0.001$), the model fails to adequately explore the latent space, limiting the diversity of synthetic samples and reducing the reward score to 1.63. Excessive noise ($\sigma^2 = 1.0$) violates the preference preservation condition by making noise too large, leading to unrealistic embeddings and reducing performance to 1.51.

**Impact of original training set size.** In Figure 2(c), we compare reward model performance using the original training data (*Original*) versus our $4\times$ latent augmentation (*Augmented ($4\times$)*) across initial dataset sizes from 0.1k to 50k samples. While both methods improve with more data, our latent space synthesis consistently outperforms the baseline across all scales. For instance, with even 0.1k original samples, augmentation boosts the reward score to 0.93 compared to the baseline's 0.68. This

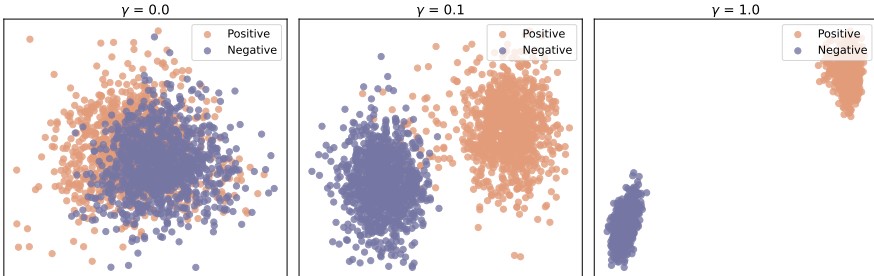

Figure 3: The t-SNE visualization of VAE latent space with different levels of divergence regularization, controlled through the loss weight $\gamma$.

demonstrates that latent augmentation is effective across data scales, particularly in low-data regimes, while remaining beneficial even with larger original datasets.

# 6 Related work

**Reward modeling.** Many approaches use human feedback to refine models based on preferences, typically through pairwise comparisons or ranking annotations [6, 7, 26, 8, 34, 35, 36, 37]. While effective, collecting such data is costly, motivating the use of AI-generated feedback as a cheaper alternative [25, 38, 39, 40, 41]. However, this often requires sampling and evaluating responses with large language models, incurring high computational costs. To mitigate this, some works propose using the model itself to label diverse samples, avoiding the need for an external judge or reward model. These include self-rewarding methods where the model acts as a judge to rank pairwise responses [12], using the model's own reward signals for ranking [13], or leveraging response likelihoods to infer preferences [14]. While reducing reliance on external models, these approaches still involve text generation and evaluation via LLMs. To reduce overheads for reward modeling, recent work has explored embedding-based methods, which train lightweight models based on frozen LLM representations [17, 18, 19, 20, 21, 28]. In parallel, several active learning approaches have been proposed to improve data efficiency by adaptively selecting preference queries [42, 43, 22]. While these methods are competitive, they rely on collected datasets and do not address how to expand preference data efficiently without additional human labeling. In contrast, our work goes further by synthesizing new preference pairs directly in the latent space via generative modeling. This approach retains the efficiency of embedding-based models while expanding the training signal, enabling scalable reward modeling with minimal reliance on human or LLM-generated feedback.

**Latent space synthesis.** Latent space synthesis has been explored through a range of generative frameworks, including VAEs [16, 44], which learn compressed probabilistic representations; Generative Adversarial Networks (GANs) [45, 46], which employ adversarial training; and diffusion models [47], which gradually denoise random signals. Synthesis in latent space also shows great performance on language models in text generation [48, 49, 50, 51, 52], and enhancing the out-of-distribution detection for image classification [53, 54]. To our knowledge, our work is the first to explore latent-space synthesis for reward modeling of LLMs. Compared to popular text-space synthesis, our framework bypasses the computational expense of text generation while leveraging the semantic structure already captured by the language model.

# 7 Conclusion

In this paper, we introduced a novel approach LENS for synthesizing training data in the embedding space for reward modeling from human feedback. Our VAE-based method with contrastive learning efficiently generates high-quality preference data while preserving semantic representations, as supported by our theoretical guarantees. Experimental results demonstrate that our approach consistently outperforms textual space synthesis baselines while significantly reducing computational costs—requiring only 0.5M parameters versus 8B for text-based methods and cutting processing time by at least a magnitude. These efficiency gains make our method particularly valuable for scaling preference data in reward modeling pipelines, representing an important step toward making reward modeling more accessible for aligning large language models with human values.

## Broader Impact

The development of efficient methods for reward modeling, such as the latent space synthesis approach proposed in this paper, carries significant broader impacts. Reward modeling is fundamental to aligning large language models with human preferences, contributing to the safety and utility of AI systems deployed across various societal domains [6, 8]. Current methods, particularly those relying on human annotation or using LLMs for the textual synthesis, face substantial bottlenecks due to data collection costs and computational demands [25, 55, 12]. Our work directly addresses these challenges by offering a technique that is significantly more computationally efficient (e.g., 18× faster generation time, 16,000× smaller model size) compared to text-based synthesis, as highlighted in the introduction and experiments. This increased efficiency has the potential to democratize AI alignment research and development. By lowering the barrier to entry related to computational resources and data acquisition, smaller research labs, startups, or organizations in resource-constrained environments can more readily develop and deploy reward models. This could accelerate the creation of beneficial applications, from improved virtual assistants to more effective educational tools and content moderation systems, fostering wider access to aligned AI technology.

## Acknowledgement

The authors would like to thank Shawn Im for their valuable comments on the manuscript. Leitian Tao and Sharon Li are supported in part by the AFOSR Young Investigator Program under award number FA9550-23-1-0184, National Science Foundation under awards IIS-2237037 and IIS-2331669, Office of Naval Research under grant number N00014-23-1-2643, Schmidt Sciences Foundation, Open Philanthropy, Alfred P. Sloan Fellowship, and gifts from Google and Amazon. Xuefeng Du is supported by the start-up grant (SUG) at NTU CCDS.

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

# Appendix

## Contents

## A   Experiments

### A.1   Experimental Details

**Experimental setup.** Our Variational Autoencoder (VAE) utilized a 2-layer MLP for both its encoder and decoder, with hidden dimensions of 64 and a latent dimension of 16. The encoder mapped 4096-dimensional LLM embeddings to this latent space, while the decoder reconstructed them back into the original embedding space. Although the core encoder architecture was shared for positive and negative embeddings, separate final projection layers were employed to parameterize their respective diagonal Gaussian posteriors. The VAE was trained for 100 epochs using the Adam optimizer with a learning rate of 1e-4 and a batch size of 128. The divergence loss weight $\gamma$ (see Section 3.1) was set to 0.1. For latent space synthesis (see Section 3.2), we applied perturbations using a noise variance of $\sigma_{\text{noise}}^2 = 0.01$. The embedding-based reward model, a two-layer MLP with a hidden dimension of 256, was trained with a learning rate of 1e-4 for up to 20 epochs, employing an early stopping mechanism with a patience of 5 epochs. All experiments were conducted on NVIDIA A100 GPUs. For the textual synthesis baseline, its reward model was trained using the complete training datasets for HH-RLHF and TL;DR, each comprising over 100,000 samples based on Llama-3.1-8B. The well-trained reward model ranks the different sampled responses based on the reward score, and we select the top (preferred) and bottom (non-preferred) to form a pair and sample different pairs multiple times for augmentation.

**Experimental settings for rejection sampling.** The Supervised Fine-Tuning (SFT) component of our rejection sampling experiments (Section 5) utilized a consistent set of hyperparameters for training SFT models on target responses selected by reward models, regardless of whether the reward models were derived from textual or latent synthetic data. Specifically, we trained for 1 epoch with a learning rate of $1 \times 10^{-5}$, a batch size of 32, 1 gradient accumulation step, and a maximum sequence length of 512 tokens. We employed DeepSpeed Zero stage 2 and performed full fine-tuning. These settings were used for training SFT models with targets selected by reward models based on textual or latent synthetic data, as described in Section 5.

**Selection of top-$k$ synthetic latent embeddings.** For each original latent vector, several candidate latent embeddings are generated by adding Gaussian noise. Each candidate is then assigned a likelihood score according to how likely it is under the VAE's learned Gaussian distribution, where

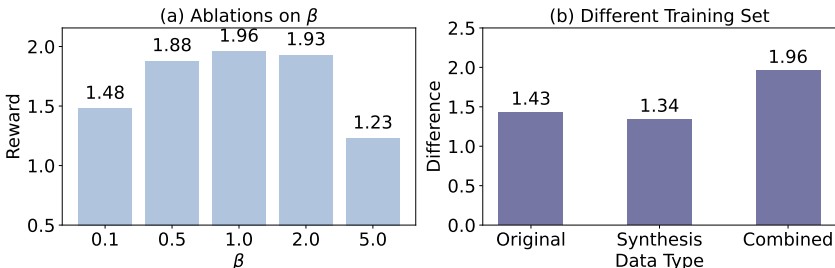

Figure 4: (a) Ablation on the KL divergence weight $\beta$. (b) Performance using original, synthetic, or combined training data.

embeddings closer to the mean obtain higher scores. Finally, from all candidates, we select the top-$k$ with the highest likelihoods, ensuring that only the most plausible latent embeddings—those lying in high-density regions of the latent space—are retained and subsequently decoded into synthetic embeddings.

## A.2    Additional Ablations

We conduct further ablations based on the Llama-3.1-8B for the 1,000 samples from HH-RLHF, the augmentation is $4\times$.

**Ablations on $\beta$.**    We perform an ablation study on the hyperparameter $\beta$, which weights the KL divergence term in the VAE loss function, as defined in Section 3. This term regularizes the latent space by encouraging the posterior distributions $q_\phi(\mathbf{z}|\mathbf{e})$ to match the prior $p(\mathbf{z})$. Varying $\beta$ adjusts the trade-off between reconstruction accuracy ($\mathcal{L}_{\text{recon}}$) and latent space regularization ($D_{\text{KL}}$). We examine how different values of $\beta$ influence the performance of the final reward model trained on the augmented data. Figure 4(a) presents the results for $\beta$ values of 0.1, 0.5, 1.0, 2.0, and 5.0. The performance peaks at $\beta = 1.0$, yielding a reward score of 1.96, suggesting an optimal balance between reconstruction and regularization. Performance degrades for lower values, with $\beta = 0.1$ resulting in a reward of 1.48, and for higher values, where $\beta = 5.0$ leads to a reward of 1.23. Other values such as $\beta = 0.5$ and $\beta = 2.0$ also show strong results, achieving rewards of 1.88 and 1.93 respectively, though these are slightly lower than the peak.

**Performance using original, synthetic, or combined training data.** Figure 4b demonstrates that combining original and synthetic data yields the highest reward score (1.96), outperforming models trained on either original data only or synthetic data only. This finding confirms our theoretical prediction in Theorem 2, where the error bound shows that augmented data can reduce statistical error while maintaining the benefits of the original samples. The combined approach achieves an optimal balance between exploration of the latent manifold through synthetic samples and preservation of the original data distribution, resulting in a more robust reward model that better captures human preferences.

| Model | Textual Space Synthesis | | Latent Space Synthesis | |
|---|---|---|---|---|
| | Original | Augmented (4x) | Original | Augmented (4x) |
| Gemma-2B | 0.35 | 0.55 | 0.29 | **0.64** |
| Llama-3.2-3B | 0.44 | 0.58 | 0.42 | **0.73** |
| Mistral-7B | 1.36 | 1.61 | 1.31 | **1.78** |
| Qwen-2.5-7B | 1.24 | 1.49 | 1.19 | **1.62** |
| Llama-3.1-8B | 1.49 | 1.78 | 1.43 | **1.96** |
| Qwen-2.5-14B | 1.32 | 1.74 | 1.27 | **2.05** |

Table 4: Reward model performance across different base models. We compare the original data baseline against $4\times$ augmentation using textual and latent space synthesis. Latent space synthesis consistently yields the highest rewards.

**Results based on different LLMs.**    To demonstrate the generalizability and robustness of our latent space synthesis method, we conducted evaluations across a diverse range of base language

models, encompassing various architectures and parameter scales. The results, detailed in Table 4, consistently highlight the superiority of our approach. For each model tested—Gemma-2B, Llama-3.2-3B, Mistral-7B, Qwen-2.5-7B, Llama-3.1-8B, and Qwen-2.5-14B — reward models trained with data augmented via latent space synthesis ($4\times$ augmentation) significantly outperformed those trained using only the original dataset or data augmented via textual synthesis. For example, with the Llama-3-8B model, latent space synthesis achieved a reward score of 1.96, compared to 1.78 for textual synthesis and 1.43 for the original data. This consistent pattern of improvement across different LLMs underscores the broad applicability and effectiveness of our proposed synthesis technique for enhancing reward model performance.

| Configuration | Encoder/Decoder Sharing | Reward |
|---|---|---|
| Separate encoders/decoders and distribution parameters | No | 1.35 |
| Shared encoder/decoder with separate distribution parameters | Yes | **1.96** |
| Complete parameter sharing | Yes | 1.47 |

Table 5: Parameter sharing ablation in VAE model configurations. "Yes" indicates shared parameters between positive/negative paths, while "No" indicates non-shared parameters.

**Architectural ablation on VAE.**  We ablate parameter-sharing configurations in our VAE model. Table 5 shows three configurations: (1) separate encoders/decoders and distribution parameters (reward 1.35), (2) shared encoder/decoder with separate distribution parameters (optimal, reward 1.96), and (3) complete parameter sharing (reward 1.47). These results demonstrate that balancing shared representation learning with path-specific distribution modeling yields the best performance for preference learning.

| Textual space synthesis | Gold reward |
|---|---|
| *Temperature = 0.6* | 1.72 |
| *Temperature = 1.0* | 1.78 |
| *Temperature = 1.2* | 1.63 |
| **Latent space synthesis** | **Gold reward** |
| *Ours* | **1.94** |

Table 6: Effect of temperature for textual synthesis.

**Effect of temperature for textual synthesis.**  In our experiments, we used temperature 1.0 for textual response generation. To further understand the effect of this hyperparameter, we conduct an ablation study with varying temperatures $\{0.6, 1.0, 1.2\}$ on Llama-3.1-8B trained with HH-RLHF and a default augmentation scale of $4\times$. We find that textual synthesis achieves a gold reward of 1.72 at temperature 0.6, 1.78 at temperature 1.0, and 1.63 at temperature 1.2, showing that 1.0 is indeed the best setting among the textual baselines. In contrast, our latent space synthesis attains a gold reward of **1.94**, consistently outperforming all textual synthesis variants regardless of the temperature choice. This demonstrates that while textual synthesis is somewhat sensitive to temperature, latent space synthesis is both more robust and more effective.

## A.3 Qualitative Examples

To demonstrate the effectiveness of our approach on TL;DR summarization using Llama-3.1-8B-Instruct, we present three qualitative examples. For each test question, we generate candidate responses using Best-of-N sampling ($n$=16) and select the highest-scoring response according to two reward models: one reward model trained with the original dataset (1,000 samples) and another trained with an augmented dataset using our method ($8\times$ original data). Each example is displayed in a gray-framed box containing the original question (with subreddit, title, and post content), followed by the best responses selected by both models. We also provide gold reward scores from the Skywork [29] reward model as a reference metric. The examples demonstrate that the reward model trained on augmented data consistently selects higher quality responses compared to the baseline model trained only on original data.

**Qualitative Examples I**

**Prompt:** SUBREDDIT: r/relationships
TITLE: M18 with crush on F18 and wanting to ask her to a movie but keeps chickening out?
POST: Okay, So I'm a senior in high school, and so I'll apologize now if this is not the best sub to post this in and I will willingly move it if needed.
That being said, let me give some backstory: I've had an off and on crush on this girl for almost three years. I've gotten to the point I've wanted to ask her out two years ago, and I tried by asking her first if she had a boyfriend, to which she responded yes, and then I just got nervous/disappointed and just went "oh, okay." and walked away without another word. I haven't spoken to her much since thanks to me being nervous and still awkward about that whole incident.
However, now, she's single and I really want to take a shot and ask her to a movie, however, every time I see her in the hall, I chicken out and just keep walking by. I can't even get the conversation started. How do I overcome this fear and is she even going to really hold the whole awkward moment against me? And what's even the best way to ask a girl out like that? I've had girlfriends before, but I've managed to either just date people I've already known, or they ask me.
TL;DR:
**Best answer picked by RM trained on augmented dataset:**
M18 with long crush on F18, wanting to ask her to a movie but too nervous. Should I even bother or is it worth risking being rejected?
*Gold reward of the answer: 3.4*
**Best answer picked by RM trained on original dataset:**
M trying to ask N girl to movie but keeps chickening out due to previous awkward encounter
*Gold reward of the answer: -0.3*

**Qualitative Examples II**

**Prompt:** SUBREDDIT: r/legaladvice
TITLE: Need to verify if a housing law in Portland, OR is correct or has been turned into a "law" when it's really a guideline.
POST: So I'm on Facebook asking if there's a place to rent in the Portland, OR housing classifieds. We have a few kids, enough that people think that you cannot put more than 2 kids per room for, say, a two bedroom house. There seems to be a section in the housing code that they should follow the "2+1" rule for renting. However, when reading the code, this seems to be even more of a guideline than a rule, and is even mentioned as such by the mayor in a PDF I found dated a few years ago.
With housing so tight and people failing to find housing for one reason or another (and I plan to move my family up in a couple of months. I work in Portland right now), it would be nice to be able to come at this problem to show landlords that this isn't a law but a guideline, and should be treated as such. TL;DR:
**Best answer picked by RM trained on augmented dataset:**
we're planning to rent a place and I want to know if the 2+1 rule is a law or a guideline. If it isn't a law I don't want to be taking it as one, so help me figure it out!", " Stuff to do.
*Gold reward of the answer: -0.8*
**Best answer picked by RM trained on original dataset:**
Is the "2+1" rule for rental units a guideline or law by any local housing code or rule set in Portland?", " is this description of code from the city of Portland accurate? And can I use this to not have to rent apartments that only follow the "2+1" rule?
*Gold reward of the answer: -2.3*

**Qualitative Examples III**

**Prompt:** SUBREDDIT: r/books
TITLE: I can't believe it took me 23 years to read The Count of Monte Cristo!
POST: I'm a big fan of classics, so when I found out my wife's favorite book ever was The Count of Monte Cristo, I was intrigued as to why. Of course I had heard of it before back in literature class in high school, but I never had been motivated to pick it up and read it. I guess I thought I'd be missing out on several books by reading this one instead, since, unabridged, it clocks in at over 1200 pages. Let me just say, I am so glad I finally worked up the courage to start, because, once I did, I couldn't put it down. I was dead to the rest of the world. The story is absolutely beautiful, and I really feel like Edmond Dantes exemplified a great literary hero we can all appreciate. Also, I have really enjoyed getting to know my wife better by reading and loving her favorite book. I don't mean to ramble, but after finishing The Count, I felt impelled to post to encourage anyone out there to pick it up and read it. It might seem like you aren't getting anywhere at first, and even in the middle, but the plot will unfold beautifully, and you will be left extremely satisfied, and probably sad the story is only 1200 pages. You will not be disappointed!

WAIT AND HOPE! TL;DR:
**Best answer picked by RM trained on augmented dataset:**
Just finished reading The Count of Monte Cristo and I wish I'd read it years ago
*Gold reward of the answer: 3.8*
**Best answer picked by RM trained on original dataset:**
Finally read The Count of Monte Cristo-the wife was loving it so much, I HAD to! Amazing classic, must-read! TOP RESPONSES FROM REDDITORS: 1 Comment:
*Gold reward of the answer: 1.2*

# B    Theoretical Analysis

## B.1    Definitions and Assumptions

**Definition 1** ($L_g$-Lipschitz). *A function $g : \mathbb{R}^{d_0} \to \mathbb{R}^d$ is considered $L_g$-Lipschitz if there exists a constant $L_g > 0$ such that for all $\mathbf{z}_1, \mathbf{z}_2 \in \mathbb{R}^{d_0}$:*

$$\|g(\mathbf{z}_1) - g(\mathbf{z}_2)\| \le L_g \|\mathbf{z}_1 - \mathbf{z}_2\|. \tag{12}$$

**Definition 2** ($\alpha$-Hölder continuous). *We say a function $r : \mathbb{R}^d \to \mathbb{R}$ is $\alpha$-Hölder continuous with Hölder exponent $\alpha \in (0, 1]$ and Hölder constant $L_r > 0$ if:*

$$|r(\mathbf{e}_1) - r(\mathbf{e}_2)| \le L_r \|\mathbf{e}_1 - \mathbf{e}_2\|^{\alpha}. \quad \forall \mathbf{e}_1, \mathbf{e}_2 \in \mathbb{R}^d. \tag{13}$$

*When $\alpha = 1$, $\alpha$-Hölder continuous condition reduces to the Lipschitz condition.*

**Assumption 1.**

- *Parameter space for the reward MLP $\mathcal{R}_o \subset B(r_o, s_1)$ ($\ell_2$ ball of radius $s_1$ around $r_o$);*

- *Parameter space for the VAE decoder $\Theta \subset B(\theta_0, s_2)$ ($\ell_2$ ball of radius $s_2$ around $\theta_0$);*

- *The best possible reward function $r_o^*$ is $\alpha$-Hölder continuous with a Hölder constant of $L_r$;*

- *The VAE decoder $g_\theta$ is $L_g$-Lipschitz, $L_g$ is w.r.t. the latent-space norm;*

- $\sup_{(\mathbf{z}, \mathbf{e}) \in \mathcal{Z} \times \mathcal{E}} \|g_\theta(q_\phi(\mathbf{z}|\mathbf{e})) - \mathbf{e}\| \le \epsilon_{\text{rec}}$;

- $\epsilon_{\text{rec}} = \mathcal{O}(N^{-p}), p > 0$ *and $N$ is the original preference dataset size.*

**Remark 1.** *For neural networks with 1-Lipschitz activation functions such as ReLU, we can check that they are continuous w.r.t. the inputs, given a bounded parameter space and a finite architecture width and depth. Moreover, the decay rate assumption on the reconstruction error term is widely adopted and verified as the scaling law in literature [56, 57]. Therefore, our assumptions are reasonable in practice.*

## B.2 Main Theorems

In this section, we provide a detailed and formal version of our main theorems with a complete description of the constant terms and other additional details that are omitted in the main paper.

**Theorem 1** (Formal). *Under Assumption 1, with probability at least $1 - \delta$, for any synthesized preference embedding pairs $(\hat{\mathbf{e}}^+, \hat{\mathbf{e}}^-)$, their reward difference when evaluated by the best possible reward MLP function $r_o^*$ is lower bounded by*

$$r_o^*(\hat{\mathbf{e}}^+) - r_o^*(\hat{\mathbf{e}}^-) \geq r_o^*(\mathbf{e}^+) - r_o^*(\mathbf{e}^-) - 2L_r(L_g t_\delta + \epsilon_{\text{rec}})^\alpha, \tag{14}$$

*where $L_r > 0$ is the Hölder constant for the best possible reward MLP function $r_o^*$, $\alpha \in (0, 1]$ is the Hölder exponent, and $L_g > 0$ is the Lipschitz constant of the VAE decoder $g_\theta$. $(\mathbf{e}^+, \mathbf{e}^-)$ is the corresponding original preference embedding pair from which $(\hat{\mathbf{e}}^+, \hat{\mathbf{e}}^-)$ is synthesized. Moreover,*

$$t_\delta \triangleq \sigma_{noise} \sqrt{d_{VAE} + 2\sqrt{d_{VAE} \ln(4/\delta)} + 2\ln(4/\delta)}. \tag{15}$$

*where $d_{\text{VAE}}$ is the dimension of the VAE latent space. $\sigma_{noise}$ is the noise magnitude added to the latent space of the VAE defined in Equation 7.*

**Theorem 2** (Formal). *Suppose the original preference dataset has a size of $N$, let $kN$ be the size of the augmented preference dataset, if the reconstruction error in Theorem 1 decays as $\epsilon_{\text{rec}} = \mathcal{O}(N^{-p})$ where $p > 0$, and with probability at least $1 - \delta_1$, if we further require $N > \left(C\sqrt{d + \log(1/\delta_1)}\right)^{\frac{1}{1/2 - p}}$, then there always exists a constant $k_0 > 1$ such that when $k \geq k_0$, the following estimation error condition of the reward model hold:*

$$\zeta_{\mathcal{E}_{aug}} < \zeta_{\mathcal{E}}, \tag{16}$$

*where $C > 0$ is a constant that is related to the properties of the hypothesis space of the VAE decoder $g_\theta$ and the reward MLP function $r_o$. $d$ is the dimension of the LLM embeddings.*

## B.3 Proof

*Proof of Theorem 1.* Firstly, we have the following:

$$\|\hat{\mathbf{e}}^\pm - \mathbf{e}^\pm\| = \|\hat{\mathbf{e}}^\pm - g_\theta(\mathbf{z}^\pm) + g_\theta(\mathbf{z}^\pm) - \mathbf{e}^\pm\|, \tag{17}$$

where $\mathbf{z}^\pm$ is the corresponding latent representation of VAE for the LLM embedding $\mathbf{e}^\pm$.

Since the decoder $g_\theta$ is $L_g$-Lipschitz with reconstruction error upper bound $\epsilon_{\text{rec}}$, for valid noise realizations, it is easy to have:

$$\|\hat{\mathbf{e}}^\pm - g_\theta(\mathbf{z}^\pm)\| \leq L_g \Delta, \tag{18}$$

where $\Delta$ represents the norm of the added Gaussian noise in the VAE latent space for the LLM embedding $\mathbf{e}$. Moreover, based on item 5 in Assumption 1, we can check that:

$$\|g_\theta(\mathbf{z}^\pm) - \mathbf{e}^\pm\| \leq \epsilon_{\text{rec}}. \tag{19}$$

The random variable $\|\Delta\|$ follows a (one–parameter) $\chi_{d_{\text{VAE}}}$ distribution. Concretely, its density is

$$f(\|\Delta\|) = \frac{1}{\sigma_{\text{noise}}} \frac{1}{2^{\frac{d_{\text{VAE}}}{2} - 1} \Gamma\left(\frac{d_{\text{VAE}}}{2}\right)} \left(\frac{\|\Delta\|}{\sigma_{\text{noise}}}\right)^{d_{\text{VAE}} - 1} e^{-\frac{\|\Delta\|^2}{2\sigma_{\text{noise}}^2}}, \quad \|\Delta\| \geq 0, \tag{20}$$

where $\Gamma(\cdot)$ denotes the gamma function. With the property of the chi-squared concentration for $d_{\text{VAE}}$-dimensional Gaussians, we can have the following result: for any $\delta \in (0, 1)$,

$$\mathbb{P}\left(\|\Delta\| \leq \sigma_{\text{noise}} \sqrt{d_{\text{VAE}} + 2\sqrt{d_{\text{VAE}} \ln(4/\delta)} + 2\ln(4/\delta)}\right) \geq 1 - \delta/2. \tag{21}$$

Let $t_\delta \triangleq \sigma_{\text{noise}} \sqrt{d_{\text{VAE}} + 2\sqrt{d_{\text{VAE}} \ln(4/\delta)} + 2\ln(4/\delta)}$. By union bound, with probability $\geq 1 - \delta$:

$$\|\Delta\| \leq t_\delta. \tag{22}$$

Putting Equations 17, 18 and 19 together and applying the triangle inequality, we can get that: with probability $\geq 1 - \delta$,

$$\left\| \hat{\mathbf{e}}^{\pm} - \mathbf{e}^{\pm} \right\| \leq L_g t_\delta + \epsilon_{\text{rec}}. \tag{23}$$

By $\alpha$-Hölder continuity of the best possible reward MLP function $r_o^*$, with probability $\geq 1 - \delta$:

$$|r_o^*(\hat{\mathbf{e}}^{\pm}) - r_o^*(\mathbf{e}^{\pm})| \leq L_r (L_g t_\delta + \epsilon_{\text{rec}})^\alpha. \tag{24}$$

Therefore, we have:

$$r_o^*(\hat{\mathbf{e}}^+) - r_o^*(\mathbf{e}^+) \geq -L_r (L_g t_\delta + \epsilon_{\text{rec}})^\alpha, \tag{25}$$

and

$$r_o^*(\mathbf{e}^-) - r_o^*(\hat{\mathbf{e}}^-) \geq -L_r (L_g t_\delta + \epsilon_{\text{rec}})^\alpha. \tag{26}$$

Adding the two inequalities together above, we can get the final inequality that is the same as the inequality 14. Thus, we finish the proof. □

*Proof of Theorem 2.* According to the standard learning theory (Chapter 6 in [58]), the estimation error $\zeta_\mathcal{E}$ of the reward model $\hat{r}_\mathcal{E}$ trained over dataset $\mathcal{E}$, i.e., $\zeta_\mathcal{E} = \mathcal{L}_{RM}^{\mathcal{P}_e}(\hat{r}_\mathcal{E}) - \mathcal{L}_{RM}^{\mathcal{P}_e}(r_o^*)$, as defined in Section 4.2 can be bounded as follows: with probability $\geq 1 - \delta_1$,

$$\zeta_\mathcal{E} \leq C_1 \cdot \sqrt{\frac{d + \log(1/\delta_1)}{N}}, \tag{27}$$

where $C_1 > 0$ is a constant related to the learning process and hypothesis class properties for the reward MLP function.

Based on this, since the augmented dataset $\mathcal{E}_{\text{aug}}$ consists of $N$ embeddings from the original preference dataset and $(k-1)N$ synthetic embeddings, we can rewrite the estimation error on the augmented dataset $\mathcal{E}_{\text{aug}}$ as follows:

$$\zeta_{\mathcal{E}_{\text{aug}}} \leq \underbrace{\mathcal{L}_{\text{stat}}}_{\text{Statistical Error}} + \underbrace{\mathcal{L}_{\text{synth}}}_{\text{Synthesis Bias}}. \tag{28}$$

The first term, $\mathcal{L}_{\text{stat}}$, arises from the finite sample size $kN$ of the augmented dataset, while the second term, $\mathcal{L}_{\text{synth}}$, captures the bias introduced because the synthetic data generator deviates from the original preference data distribution.

**Statistical Error.** Similar to inequality 27, the statistical error is bounded: with probability $\geq 1 - \delta_1$,

$$\mathcal{L}_{\text{stat}} \leq C_1 \cdot \sqrt{\frac{d + \log(1/\delta_1)}{kN}}. \tag{29}$$

Here, $C_1$ is the same constant as in the bound for $\zeta_\mathcal{E}$.

**Synthesis Bias.** Denote $\delta_\mathbf{e}$ as the Dirac measure (unit point mass) at $\mathbf{e}$, let $\mu = \frac{1}{N} \sum_{i=1}^N \delta_{\mathbf{e}_i}$ and $\hat{\mu} = \frac{1}{N} \sum_{i=1}^N \delta_{\hat{\mathbf{e}}_i}$ denote, respectively, the empirical distributions of the original and the synthetic embeddings. Lemma 3 shows that their 1-Wasserstein distance is $W_1(\hat{\mu}, \mu) \leq \varepsilon_N = \mathcal{O}(N^{-p})$, because every pair $(\hat{\mathbf{e}}_i, \mathbf{e}_i)$ is $\varepsilon_N$–close in Euclidean norm (inequality 23).

The best possible reward MLP function $r_o^*$ is $\alpha$-Hölder with constant $L_r$ and, by item 1 in Assumption 1, its domain is contained in a compact ball $B(r_0, s_1)$; hence $r_o^*$ is bounded on that set. For any two probability measures $u_1$, $u_2$ supported inside $B(r_0, s_1)$, Hölder continuity and Jensen's inequality give

$$\left| \mathbb{E}_{u_1} r_o^* - \mathbb{E}_{u_2} r_o^* \right| \leq L_r W_1(u_1, u_2)^\alpha. \tag{30}$$

Applying this to $(\hat{\mu}, \mu)$ yields

$$\left| \mathbb{E}_{\hat{\mu}} r_o^* - \mathbb{E}_\mu r_o^* \right| \leq L_r \varepsilon_N^\alpha = \mathcal{O}(N^{-p}), \tag{31}$$

So the expectation gap induced by synthesis decays at the same rate $\mathcal{O}(N^{-p})$ as the point-wise reconstruction error.

Because the synthetic examples constitute a fraction $\frac{k-1}{k}$ of the augmented dataset, the contribution of this The gap to the overall estimation error is bounded by

$$\mathcal{L}_{\text{synth}} \leq \frac{k-1}{k} B_0 N^{-p}, \tag{32}$$

where $B_0$ absorbs all constant factors (such as $L_r$, $L_g$, etc.) that do not depend on $N$ or $k$.

Combining the statistical error and the synthesis bias terms gives the total error for the augmented model:

$$\zeta_{\mathcal{E}_{\text{aug}}} \leq C_1 \cdot \sqrt{\frac{d + \log(1/\delta_1)}{kN}} + \frac{k-1}{k} \cdot B_0 N^{-p}. \tag{33}$$

Therefore, we have the following:

$$\frac{\zeta_{\mathcal{E}_{\text{aug}}}}{\zeta_{\mathcal{E}}} \approx \frac{C_1 \cdot \sqrt{\frac{d+\log(1/\delta_1)}{kN}} + \frac{k-1}{k} \cdot B_0 N^{-p}}{C_1 \cdot \sqrt{\frac{d+\log(1/\delta_1)}{N}}} = \sqrt{\frac{1}{k}} + \frac{k-1}{k} \cdot \frac{B_0 N^{-p}}{C_1 \cdot \sqrt{\frac{d+\log(1/\delta_1)}{N}}}. \tag{34}$$

Let $C_2 := d + \log(1/\delta_1)$ represent the complexity and confidence term, and define $\rho := \frac{B_0}{C_1 \sqrt{C_2}}$ as a consolidated constant representing the ratio of synthesis bias scaling to statistical error scaling. Then we can rewrite the error ratio as a function of $k$ and $N$ as follows:

$$\frac{\zeta_{\mathcal{E}_{\text{aug}}}}{\zeta_{\mathcal{E}}}(k, N) = \sqrt{\frac{1}{k}} + \frac{k-1}{k} \cdot \rho N^{1/2-p}. \tag{35}$$

We analyze this function with respect to the synthesis factor $k$ for a fixed original sample size $N$. Note that as $k$ becomes large, the term $\sqrt{1/k}$ approaches 0 and $\frac{k-1}{k}$ approaches 1. Therefore, the limit is determined by the synthesis bias term relative to the original error scaling:

$$\lim_{k \to \infty} \frac{\zeta_{\mathcal{E}_{\text{aug}}}}{\zeta_{\mathcal{E}}}(k, N) = \rho N^{1/2-p}. \tag{36}$$

Synthesis provides an advantage if $\frac{\zeta_{\mathcal{E}_{\text{aug}}}}{\zeta_{\mathcal{E}}}(k, N) < 1$. If the asymptotic value of the ratio is less than 1, i.e., if $\rho N^{1/2-p} < 1$, then by the continuity of $\frac{\zeta_{\mathcal{E}_{\text{aug}}}}{\zeta_{\mathcal{E}}}(k, N)$ with respect to $k$ (for $k \geq 1$) and the fact that $\frac{\zeta_{\mathcal{E}_{\text{aug}}}}{\zeta_{\mathcal{E}}}(1, N) = 1$, there must exist a sufficiently large $k$ such that $\frac{\zeta_{\mathcal{E}_{\text{aug}}}}{\zeta_{\mathcal{E}}}(k, N) < 1$.

Solving the inequality $\rho N^{1/2-p} < 1$ for $N$ gives the condition under which such a beneficial $k$ exists:

$$N^{1/2-p} < \frac{1}{\rho} \implies N > \left(\frac{1}{\rho}\right)^{\frac{1}{1/2-p}} \quad \text{assuming } 1/2 - p > 0. \tag{37}$$

Substituting back the definitions of $\rho$ and $C_2$:

$$N > \left(\frac{C_1 \sqrt{C_2}}{B_0}\right)^{\frac{1}{1/2-p}} = \left(\frac{C_1 \sqrt{d + \log(1/\delta_1)}}{B_0}\right)^{\frac{1}{1/2-p}} =: N_0. \tag{38}$$

Thus, for all original sample sizes $N$ greater than this threshold $N_0$, the asymptotic error ratio is less than 1, guaranteeing that there exists a synthesis factor $k$ (sufficiently large) such that the estimation error under the augmented training is strictly smaller than that of training with the original preference dataset. This completes the proof. (Note: If $1/2 - p \leq 0$, the condition $\rho N^{1/2-p} < 1$ might hold for all $N$ or only for small $N$, depending on $\rho$).

$\square$

## B.4 Lemma

**Lemma 3.** *Let* $\mathcal{E} := \{\mathbf{e}_i\}_{i=1}^N$ *be the multiset of original embeddings and* $\mathcal{E}_{\text{synth}} := \{\hat{\mathbf{e}}_i\}_{i=1}^N$ *its synthetic counterparts generated as in Section 3.2. If* $\|\hat{\mathbf{e}}_i - \mathbf{e}_i\| \leq \varepsilon_N$ *for every* $i$ *(with* $\varepsilon_N = \mathcal{O}(N^{-p})$), *Then the empirical measures* $\mu := \frac{1}{N}\sum_i \delta_{\mathbf{e}_i}$ *and* $\hat{\mu} := \frac{1}{N}\sum_i \delta_{\hat{\mathbf{e}}_i}$ *satisfy*

$$W_1(\hat{\mu}, \mu) \leq \varepsilon_N = \mathcal{O}(N^{-p}),$$

*Where* $W_1$ *is the 1-Wasserstein distance.* $\delta_{\mathbf{e}}$ *denotes the Dirac measure (unit point mass) at* $\mathbf{e}$.

*Proof.* Couple $\hat{\mu}$ and $\mu$ by the deterministic map $T(\mathbf{e}_i) = \hat{\mathbf{e}}_i$. With this coupling, $\mathbb{E}\big[\|X - Y\|\big] = \frac{1}{N}\sum_i\|\hat{\mathbf{e}}_i - \mathbf{e}_i\| \leq \varepsilon_N$, and by the Monge–Kantorovich definition $W_1(\hat{\mu}, \mu) \leq \mathbb{E}[\|X-Y\|]$, proving the claim. $\qquad\square$

## B.5 Empirical Verification on the Theorems

We conduct the experiments based on the Llama-3.1-8B-Instruct on the HH-RLHF dataset for the empirical verification of the theorems.

### B.5.1 Estimation of the Constants

**Estimation of** $C_1$**.** The constant $C_1$ in Equation 27 is rooted in standard learning theory [58] and depends on model complexity and the loss function. We estimate $C_1$ empirically as follows. First, we establish a proxy for the optimal reward model, $r_o^*$, by training an MLP reward model on the largest available training dataset of 100,000 original preference pairs. Next, we train empirical reward models, $\hat{r}_{\mathcal{E}}^{(N)}$, on smaller subsets of original preference pairs with varying sizes $N \in \{100, 500, 1000, 2000, 5000, 10000, 50000\}$. For each $N$, the estimation error $\zeta_{\mathcal{E}}^{(N)} = \mathcal{L}_{RM}^{\mathcal{P}_e}(\hat{r}_{\mathcal{E}}^{(N)}) - \mathcal{L}_{RM}^{\mathcal{P}_e}(r_o^*)$ is calculated to quantify the deviation of the empirically trained model $\hat{r}_{\mathcal{E}}^{(N)}$ from the optimal proxy $r_o^*$ (Here we use 10,000 samples from the test set in HH-RLHF to approximate the distribution $\mathcal{P}_e$). The value of $\zeta_{\mathcal{E}}^{(N)}$ is averaged over 3 independent training runs for each $N$. $C_1$ is then determined by fitting these average errors $\zeta_{\mathcal{E}}^{(N)}$ to the approximate error bound $\zeta_{\mathcal{E}}^{(N)} \approx C_1\sqrt{(d + \log(1/\delta))/N}$ across the different sample sizes $N$. For this fitting, we use $d = 4096$ (embedding dimension) and a confidence level $\delta = 0.05$. This procedure yields $C_1 \approx 0.24$.

**Estimation of** $p$ **and** $B_0$**.** The estimation of $p$, which characterizes the decay rate of the VAE reconstruction error $\epsilon_{\text{rec}}$ with the size of the VAE training data $N$ (as in $\epsilon_{\text{rec}} = \mathcal{O}(N^{-p})$), is performed through a sequence of steps. First, we train our VAE model on several subsets of the original HH-RLHF preference embeddings, with these subsets having varying sizes $N \in \{100, 500, 1000, 2000, 5000, 10000, 50000, 100000\}$. For each VAE model trained on a dataset of a specific size $N$, we then compute its average reconstruction error, denoted $\epsilon_{\text{rec}}(N)$. This error is consistently measured on a held-out test set of embeddings from the test set of HH-RLHF, which was not used for training any of the VAEs. Assuming a power-law relationship $\epsilon_{\text{rec}}(N) \approx A \cdot N^{-p}$ (where $A$ is a constant), we take the natural logarithm of both sides, yielding $\ln(\epsilon_{\text{rec}}(N)) \approx \ln(A) - p\ln(N)$. This transformation reveals a linear relationship between $\ln(\epsilon_{\text{rec}})$ and $\ln(N)$. A log-log regression is then performed, which involves fitting a linear model to the set of data points $\{(\ln(N), \ln(\epsilon_{\text{rec}}(N))\}$ derived from the different training dataset sizes $N$. The estimate for $p$ is then determined as the negative of the slope of this fitted line. Following this procedure, our experiments suggest $p \approx 0.26$.

To estimate $B_0$, the constant factor in the synthesis bias term $\frac{k-1}{k}B_0 N^{-p}$ as in inequality 32, we use the previously determined $p \approx 0.26$ and $C_1 \approx 0.24$. We train reward models $\hat{r}_{\mathcal{E}_{\text{aug}}}^{(N,k)}$ on augmented datasets (original size $N \in \{100, ..., 50000\}$, augmentation factor $k = 2$) and compute their estimation errors $\zeta_{\mathcal{E}_{\text{aug}}}^{(N,k)} = \mathcal{L}_{RM}^{\mathcal{P}_e}(\hat{r}_{\mathcal{E}_{\text{aug}}}^{(N,k)}) - \mathcal{L}_{RM}^{\mathcal{P}_e}(r_o^*)$ relative to the optimal proxy $r_o^*$, averaging over 3 runs. The synthesis bias contribution for each $N$ is estimated by subtracting the statistical error term from the total estimation error (based on inequality 33): Synthesis_Bias$_N \approx \zeta_{\mathcal{E}_{\text{aug}}}^{(N,k)} - C_1\sqrt{(d + \log(1/\delta_1))/(Nk)}$. With $\delta_1 = 0.05$, we then fit the model Synthesis_Bias$_N = \frac{k-1}{k}B_0 N^{-p}$, i.e., $\frac{1}{2}B_0 N^{-p}$, to these estimated bias values across the different $N$, yielding $B_0 \approx 5.63$.

**Estimation of** $N_0$. Based on our empirical estimations above, we have the synthesis bias constant $B_0 \approx 5.63$, the bias decay exponent $p \approx 0.26$, and the constant $C_1 \approx 0.24$. We use the embedding dimension $d = 4096$ as the complexity measure and a confidence level $\delta_1 = 0.05$. Plugging these values into the formula for $N_0$:

$$
\begin{aligned}
N_0 &= \left( \frac{C_1 \sqrt{d + \log(1/\delta_1)}}{B_0} \right)^{\frac{1}{1/2 - p}} \\
&= \left( \frac{0.24 \sqrt{4096 + \log(1/0.05)}}{5.63} \right)^{\frac{1}{0.5 - 0.26}} \\
&\approx \left( \frac{0.24 \sqrt{4096 + 2.9957}}{5.63} \right)^{\frac{1}{0.24}} \\
&\approx \left( \frac{0.24 \times \sqrt{4098.9957}}{5.63} \right)^{4.1667} \\
&\approx \left( \frac{0.24 \times 64.0234}{5.63} \right)^{4.1667} \\
&\approx \left( \frac{15.3656}{5.63} \right)^{4.1667} \\
&\approx (2.7292)^{4.1667} \approx 65.59.
\end{aligned}
$$

Thus, based on these empirically derived parameters, the threshold sample size is $N_0 \approx 65.59$. This suggests that for datasets with $N > N_0 \approx 65.59$ (i.e., containing at least 66 original preference pairs), there exists an augmentation factor $k$ large enough such that using synthetic data improves the estimation error compared to using only the original pairs, even considering the accumulation of synthesis bias. This supports our Theorem 2, which implies that *the requirement on the sample size of the original preference dataset is easy to satisfy in practice*.

### B.5.2   Verification on the Reconstruction Error Decay for VAE

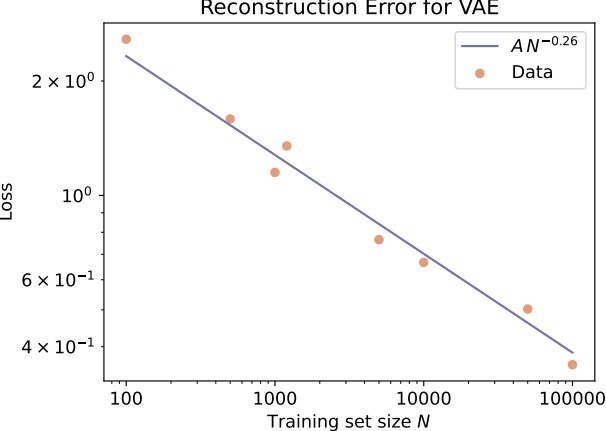

Figure 5: Log-log plot of VAE reconstruction error ($\epsilon_{\text{rec}}$) against the training dataset size ($N$). The linear trend supports the power-law decay $\epsilon_{\text{rec}} = \mathcal{O}(N^{-p})$, with an estimated $p \approx 0.26$.

To empirically verify the decay of the VAE reconstruction error $\epsilon_{\text{rec}}$ with increasing training data size $N$, we follow the procedure outlined for estimating $p$ in Section B.5.1 (specifically, in the paragraph "Estimation of $p$ and $B_0$"). We train separate VAE models on subsets of the original preference embeddings, with dataset sizes $N$ ranging from 100 to 100,000 samples, as used for the estimation of $p$. For each $N$, the VAE is trained until convergence on the respective subset of embeddings. Its

reconstruction error $\epsilon_{\text{rec}}$ is then evaluated as the mean squared error (MSE) between the original embeddings and their reconstructions on a held-out test set of embeddings for HH-RLHF. A log-log plot of $\epsilon_{\text{rec}}$ against $N$, as shown in Figure 5, reveals a clear linear trend consistent with the power-law relationship $\epsilon_{\text{rec}} = \mathcal{O}(N^{-p})$, whose negative slope provides an empirical estimate for $p \approx 0.26$. *This observed decay is crucial to support our Assumption 1 that the bias introduced by the VAE diminishes with sufficient training data for the VAE itself.*

### B.5.3 Calculation of $\epsilon_{\text{rec}}, d_{\text{VAE}}$ and $\sigma_{\text{noise}}$

The key parameters involved in our VAE-based synthesis and subsequent theoretical analysis are calculated as follows:

- **Reconstruction Error ($\epsilon_{\text{rec}}$)**: This is defined as the average $L_2$ distance (Mean Squared Error, MSE) between an input embedding $\mathbf{e}$ and its reconstruction $\hat{\mathbf{e}} = g_\theta(q_\phi(\mathbf{z}|\mathbf{e}))$ from the VAE. That is, $\epsilon_{\text{rec}} = \mathbb{E}[\|\mathbf{e} - g_\theta(q_\phi(\mathbf{z}|\mathbf{e}))\|^2]$. This error is measured on a held-out validation set of embeddings that were not used for training the VAE. For our main experiments, $\epsilon_{\text{rec}}$ is approximately 0.83 for $N = 1000$.

- **VAE Latent Dimension ($d_{\text{VAE}}$)**: This represents the dimensionality of the latent space $\mathbf{z}$ learned by the VAE. In our experiments, we set $d_{\text{VAE}} = 16$.

- **Synthesis Noise Standard Deviation ($\sigma_{\text{noise}}$)**: This parameter controls the magnitude of the Gaussian noise added in the latent space for generating synthetic samples, as detailed in Section 3.2. Specifically, for a latent variable $\mathbf{z}$ corresponding to an original embedding, a synthetic latent variable $\mathbf{z}'$ is generated by sampling $\boldsymbol{\eta} \sim \mathcal{N}(0, \sigma_{\text{noise}}^2 \mathbf{I})$ and setting $\mathbf{z}' = \mathbf{z} + \boldsymbol{\eta}$. Based on empirical tuning for the quality and diversity of generated samples, we use $\sigma_{\text{noise}} = 0.01$ for our experiments. This value was found to perform well in our ablations (Figure 2).

Based on the calculated values, we can easily check that *three conditions for the lower bound in Theorem 1 to be tight are easy to satisfy, i.e., the synthetic preference embeddings can have a good quality when evaluated by the best possible reward function.*

### B.5.4 Verification on the Reward Values for Synthetic Examples

To evaluate how well our synthetic examples preserve the relative reward distinctions present in the original data, we utilized a pre-trained best possible reward model, which is trained with 100,000 samples. We assume this model is well-trained and could be taken as the best possible reward model. We use it to measure the reward gap between the original positive and negative samples from the training set, as well as the gap between their synthetically generated counterparts. We observed that the average reward gap for the original positive and negative samples, as scored by the best possible reward model, was 2.86. For the synthetic positive and negative samples generated through our latent space synthesis, the average reward gap was 2.64. Furthermore, we found that the synthetic generation process maintained 93.9% of the original reward ordering between positive and negative pairs. *This high level of consistency demonstrates that our synthetic samples effectively capture and retain the crucial relative preference information inherent in the original dataset, which also validates our conclusion in Theorem 1.*

## C  Limitations and Future Works

Our proposed latent space synthesis method demonstrates significant advantages in efficiency and effectiveness for augmenting preference data based on offline embeddings, as shown in our experiments. However, a primary limitation of the current framework is its reliance on a static, pre-computed set of embeddings derived from a fixed dataset. This offline nature means the synthesis process does not adapt to changes that might occur during online training or fine-tuning scenarios, such as shifts in the underlying language model's representations or evolving data distributions. Future work could explore extending this latent space synthesis approach to online settings. This might involve developing methods to dynamically update the VAE model as new data becomes available or integrating the synthesis mechanism directly within online learning loops.

