# OpenReview forum: "Limited Preference Data? Learning Better Reward Model with Latent Space Synthesis"
_NeurIPS.cc/2025/Conference — NeurIPS 2025 poster_

### Official Review · Reviewer_8jXC · 2025-06-25

**Clarity:** 3
**Significance:** 3
**Originality:** 3
**Rating:** 4
**Confidence:** 3

**Summary:**

This paper proposes a novel idea to synthesize preference in the model embedding space.  It leverage well-trained VAEs to perform perturbations in the latent space, which is subsequently decoded back to the embedding space. The proposed method augments preference data at lower computational cost and demonstrates higher reward score compared to other baselines with textual synthesis.

**Questions:**

1. Does the "original" column in Table 1 represent results using only the subsampled 1,000 data points, while the 2x~8x columns indicate performance with progressively larger augmented datasets generated by the proposed method or simply sampling more responses in the text space directly?

2. The comparison methods appear to be reward modelling approaches rather than textual synthesis techniques. Could the authors explain how these baselines are directly comparable to the proposed data augmentation approach?

3. Is it possible to map the learned embeddings directly to text space and then do reward modelling?

**Ethical Concerns:**

["NO or VERY MINOR ethics concerns only"]

**Final Justification:**

My previous concerns, mainly with regard to the comparison baselines, are addressed, and I have a positive assessment of the paper.

**Limitations:**

yes

**Quality:**

3

**Strengths And Weaknesses:**

Strengths:

The paper is well-written and logically structured, with clear approach illustration as well as theoretical analysis, which makes the paper bery solid.  The authors provide thorough theoretical analysis alongside intuitive illustrations of their approach.

Weaknesses:

1. The reported results in Table 1 focus on reward model (RM) performance for HH and RLHF tasks, but the process of "textual synthesis" remains unclear. Since the latent space data synthesis is intrinsically linked to reward modeling, it's difficult to determine whether performance improvements stem from higher-quality synthesized preference data or from the MLP mapping in the reward modeling phase. A more detailed ablation study could help disentangle these effects.
2. The current comparisons primarily focus on reward modeling techniques. For a more comprehensive evaluation, the authors should include additional baselines that generate textual augmentations in the original text space, as these would provide more direct comparisons to the proposed synthesis approach.
3. Simply comparing the reward scores is limited. It would be better to see how the learned reward model performs on more generalized tasks such as on RewardBench.
4. For downstream tasks, RL algorithms such as PPO that directly utilizes reward scores can better demonstrate if the leard RM is better.

---

> ### Author Rebuttal · Authors · 2025-07-29
>
> We thank the reviewer for the constructive and helpful feedback, and for recognizing the value of our work. Below we address your concerns in detail.
>
> > W1: Clarification of the source of improvement (MLP or high-quality synthesized data?)
>
> We thank the reviewer for raising this important point. We have included the details of "textual synthesis" in **Section 5.1** and **Appendix A.1**, and are happy to elaborate on this. For each prompt in the seed set, we sample multiple responses. A reward model then ranks these responses, and we select the top (preferred) and bottom (non‑preferred) to form a pair.
>
> To disentangle whether our performance improvements stem from the quality of synthesized preference data or from the MLP reward mapping phase, we have conducted explicit ablations in Table 1. In particular, the baseline "**Embedding MLP (textual)**" uses preference pairs generated in the textual space, while fixing the feature layer and only retraining the MLP with the new samples. The only difference between this baseline and our method lies in the source of synthesized data. As shown in Table 1, our approach significantly outperforms this baseline (e.g., 1.94 vs. 1.62 reward score under 4× augmentation), which demonstrates that **the gains primarily come from the higher-quality synthesized samples rather than from the MLP**.
>
> We further examined this question by ablating simpler latent synthesis methods in Table 1: (1) Direct Noise Perturbation, where Gaussian noise is directly added to embeddings, and (2) Gaussian Sampling, where embeddings are sampled from a multivariate Gaussian fit. Both variants showed almost no improvement compared to the baseline, indicating that our VAE-based synthesis is crucial for producing semantically meaningful and diverse samples. Together, these ablations confirm that the performance improvements in Table 1 are indeed driven by the high quality of latent-synthesized preference data.
>
>
> > W2/Q2: Additional baselines that generate textual augmentations in the original text space
>
>
> We agree that it is important to compare against strong textual synthesis baselines. In fact, **our experiments in Table 1 already include three representative text-space augmentation methods. These methods differ in how the preference data is generated, rather than focusing on the reward modeling approach itself**.
>
> * Self-rewarding \[10]: Prompt the model itself as a judge to rank pairwise responses generated in the text space.
> * Self-evolved \[11]: Relies on reward-based ranking of diverse responses, also performed directly in the text space.
> * IPO \[12]: Determines preference rankings by comparing the likelihoods of generated responses, another text-based augmentation strategy.
>
> These baselines involve generating multiple textual responses for each prompt and applying ranking or selection strategies to construct preference pairs. As shown in Table 1, our latent-space synthesis method consistently outperforms these text-space augmentation baselines, demonstrating that our approach is more effective despite operating entirely in the embedding space, which is far more computationally efficient.
>
>
> > W3: comparison on the RewardBench.
>
> Thank you for your suggestion! We have evaluated the learned reward model on RewardBench. Our reward model trained with latent-space synthesis achieves competitive accuracy on the RewardBench safety subset (the goal of our alignment training) compared to models trained on textual augmentations, demonstrating that the performance gains extend to out-of-distribution preference tasks. We will include these results in the revised version. The reported results are based on Llama-3-8B on HH-RLHF with 4x augmentation.
>
>
>
> | Methods | Accuracy |
> | -------- | -------- |
> | Textual space synthesis     |   66%   |
> | Lantent space synthesis     |    **69**%  |
>
>
> > W4: online training experiments with reward model.
>
> While PPO would be an ideal downstream evaluation, its training cost is prohibitively high for large models. Instead, we adopt rejection sampling as a cost-efficient alternative to approximate online training. As demonstrated in Table 3, the reward model trained with our latent synthesis notably improves SFT outcomes and achieves a 61% win rate in GPT-4 evaluations, compared to 39% for text-based baselines. Prior work (e.g., [1]) has shown that rejection sampling can approximate the benefits of PPO-style updates, making our setup both practical and representative.
>
> [1] Gao et al., Scaling laws for reward model overoptimization, ICML 2023.
>
>
> > Q1. Does the "original" column in Table 1 represent results using only the subsampled 1,000 data points, while the 2x~8x columns indicate performance with progressively larger augmented datasets generated by the proposed method or simply sampling more responses in the text space directly?
>
> Yes, the "Original" column corresponds to training the reward model using only the 1,000 seed preference pairs subsampled from the HH dataset. The subsequent 2×–8× columns show performance when the dataset is augmented to larger sizes, starting from the same 1,000 seed pairs. Importantly, **each row in the table corresponds to a different augmentation method**: textual synthesis, simple latent variants (e.g., Gaussian sampling), and our proposed latent-space synthesis.
>
> > Q2. The comparison methods appear to be reward modelling approaches rather than textual synthesis techniques. Could the authors explain how these baselines are directly comparable to the proposed data augmentation approach?
>
> We realize that the naming of some methods in Table 1 may have caused confusion. To clarify, **all baselines in Table 1 are indeed data augmentation methods** for reward modeling, not alternative reward model architectures. Please also see our response in W2 for details.
>
> > Q3: Possibility of mapping embeddings to texts
>
> We thank the reviewer for this interesting question. In principle, one could attempt to decode synthesized embeddings back into text and then apply reward modeling in the textual space. However, this would require training or fine-tuning a high-capacity decoder that can faithfully map embeddings to fluent and semantically consistent text. This is a highly non-trivial problem by itself, since LLM embeddings are not guaranteed to be invertible to unique text sequences, and imperfect decoding could easily introduce artifacts or distort the preference signal.
>
> In contrast, our approach avoids these pitfalls by operating directly in embedding space, where the synthesized representations can be seamlessly integrated into reward model training. This design choice is both efficient (no need for text decoding) and theoretically grounded: as shown in Theorem 1, our latent-space synthesis preserves preference ordering with high probability. That said, mapping embeddings back into text is an intriguing direction for interpretability and inspection, and we view this as a promising avenue for future work, complementary to the efficiency-focused contribution of our current paper.

---

> > ### Comment · Reviewer_8jXC · 2025-08-04
> >
> > Thanks for the detailed responses, and I believe my concerns have been addressed. I am happy to maintain my current positive score.

---

> > > ### Author Response · Authors · 2025-08-04
> > >
> > > Thank you so much for taking the time to read our rebuttal. We are glad that our response addresses your concerns and we appreciate your insightful comments and support!
> > >
> > > Best,
> > > Authors

---

### Official Review · Reviewer_8VC1 · 2025-06-30

**Clarity:** 4
**Significance:** 2
**Originality:** 2
**Rating:** 4
**Confidence:** 5

**Summary:**

The paper proposes a new way to train reward model under limited data. The way they do it is through taking their limited preference data and training a VAE. Subsequently they perturb the latent space and train on the embeddings.
They show in experiments that the proposed method is able to beat many exisiting baselines on two datasets.

**Questions:**

see above weakness

**Ethical Concerns:**

["NO or VERY MINOR ethics concerns only"]

**Final Justification:**

As mentioned to the authors

I am very happy with the rebuttal and would like to thank the authors for running the additional experiments. The main reason i am not raising the score further is because i still believe that a concrete comparison with active learning methods is crucial for any practical use. In a real setting, one should always be able to access some sort of new data annotation if not for user drift alone already. However, i think the above has well addressed my concerns

**Limitations:**

see above weakness

**Quality:**

2

**Strengths And Weaknesses:**

Strenghts:
- The paper proposes a method which does not require them to have human annotators which is very interesting and beneficial for fast iterations.
-  They are tackling a good problem on limited preference data which is a big problem.
- Their proposed method does seem to outperform existing methods by a big margin and this is without "gold labels"
- Their method is also substantially more computationally efficient than exisiting text based training.


Weakness:
- Could the authors please elaborate how the textual preference data was generated and whether they also used the skywork model to label that set of preference data?
- In addition, given that the authors uses the same set of prompts to generate more preference data i.e. for a single prompt there was 2 generations at training. If the authors then generated another set of 2 generations, is there a way to now construct new preference data pairs? I know this is possibly not the best but might be worth a try to not only obtain a single new pair but potentially 3 new ones.
- What temperature was used to generate the textual preference dataset? There should be an ablation to understand the effect.
- There are no standard errors on the baseline method in table 1
- How well do you think do your method compared against active learning methods, several have been proposed and i would like to see some comparisons against them as it seems directly related to limited data. I agree that your proposed method does not require any human labels at all which i find quite surprising, however given that you have already claimed you can beat baselines with human labels, this should be one of them. [1, 2, 3]
- At what stage does the method taper off? i.e. when does getting more training data and the proposed method no longer have an effect.  In the paper you show up to 50k samples but what if we go beyond, which is most of the time the case?

If the above have been address appropriately, i would be more than happy to raise my score.

[1] Active Preference Learning for Large Language Models
[2] Active Reward Modeling: Adaptive Preference Labeling for Large Language Model Alignment (also uses embeddings)
[3] PILAF: Optimal Human Preference Sampling for Reward Modeling

---

> ### Author Rebuttal · Authors · 2025-07-29
>
> We thank the reviewer for the constructive and helpful feedback, and for recognizing the potential of our work. Below we address your concerns in detail.
>
> > W1: How was the textual preference data generated?
>
> We have included the details in **Section 5.1** and **Appendix A.1**, and are happy to elaborate further on this. For each prompt in the seed set, we sample multiple responses. A reward model based on Llama-3.1-8B trained on the training split of HH-RLHF or TL;DR ranks these responses, and we select the top (preferred) and bottom (non‑preferred) to form a pair and random pair multiple times for augmentation. We deliberately avoid the Skywork RM to prevent leakage of its 'gold' preferences into the training loop, thereby preserving the integrity of downstream evaluation.
>
> > W2: Generation to not only obtain a single new pair but potentially 3 new ones.
>
> We thank the reviewer for this excellent suggestion! **Our current implementation indeed aligns with your idea**. To clarify, when generating a new pair of latent-space preference data, we effectively obtain three additional pairs by compositionally combining each synthetic pair with the original preference data. This results in up to a 4× increase in training pairs per original triplet. This strategy is captured in Equation (8):
>
> $\mathcal{E}_{\text{aug}} = \lbrace (\tilde{e}^+, \tilde{e}^-) \mid  \tilde{e}^+ \in \mathcal{E}^+ \cup \mathcal{E_synth^{+}} \; \tilde{e}^- \in \mathcal{E}^{-} \cup \mathcal{E_synth^-} \rbrace.$
>
> This compositional pairing not only increases the diversity of training signals but also amplifies the impact of each synthesized sample. We appreciate the reviewer’s idea and will include a clarifying remark in the final version to make this more explicit.
>
>
> > W3: Ablations on temperature for the textual preference data.
>
> The temperature we used for the textual response generation is 1.0. As suggested, we conducted additional ablation for your reference. We report the results based on Llama-3-8B on HH-RLHF with 4x augmentation. The experiments are based on HH-RLHF with default augmentation scale 4×. Below we report the experimental results for different temperatures {0.6,1.0,1.2} for the textual synthesis compared to ours.
>
>
> | textual space synthesis | Gold reward |
> | -------- | -------- |
> | temperature=0.6     |  1.72   |
> | temperature=1.0     |   1.78   |
> | temperature=1.2     |   1.63   |
> | **latent space synthesis** | |
> | ours |**1.94** |
>
> We find that temperature 1.0 can yield relatively strong performance for textual synthesis, and our latent space synthesis consistently outperforms all tested temperatures.
>
> > W4: Standard errors on the baseline method.
>
> We absolutely agree that reporting standard errors improves evaluation rigor. We have conducted multiple runs for the _strongest baseline_ (full fine-tune with text-based synthesis) in Table 1 and include the results below, demonstrating that the improvement of our method is statistically significant. In the final version, we will report standard errors for baselines with textual space synthesis.
>
>
> For the Llama-3-8B on HH-RLHF,
> | Methods | 2x | 4x | 8x|
> | -------- | -------- | -------- | -------- |
> |   Textual space synthesis (full fine-tune)    | 1.53 ±0.09   | 1.83 ±0.12    |1.97±0.16 |
> | Latent-space synthesis      | **1.86±0.04**     | **1.94±0.06**    | **2.20±0.12** |
>
> For the Llama-3-8B on TL;DR,
> | Methods | 2x | 4x | 8x|
> | -------- | -------- | -------- | -------- |
> |   Textual space synthesis (full fine-tune)    | 0.86±0.07   |1.04±0.12    |1.18±0.11 |
> | Latent-space synthesis      |**1.25±0.03**     |**1.44±0.05**    | **1.48±0.07** |
>
>
> > W5: Comparison with active learning methods.
>
> We thank the reviewer for bringing up active learning, which is indeed a relevant and important direction in the limited data setting. Our method addresses a complementary challenge: rather than selecting new queries for human annotation (as in active learning), we focus on augmenting an existing set of preference-labeled data **without** requiring any additional human labeling---as you correctly recognized.
>
> To clarify, we do not claim that our method outperforms baselines trained with human-labeled data. Instead, all compared methods—including ours—start from the same set of existing preferences, and differ in how they augment this base dataset. There is no human involved in the augmentation phase for all baselines. While prior work typically augments in text space (e.g., generating more responses and labeling them with a reward model), our method performs augmentation directly in embedding space, which is more efficient and yields better downstream generalization.
>
> Though direct comparison is not fair or strictly applicable, we do believe that exploring hybrid strategies—e.g., using active learning to select high-impact prompts for latent synthesis—is a promising direction for future work, and we will add this discussion and cite relevant works as you mentioned to the paper.
>
>
> > W6: What stage does the method taper off?
>
>
>
> Great question! We provide experimental results training the reward model with varying training set sizes, including sizes larger than 50K, as shown below. Even when using the full training set (~100K samples)—the largest size we were able to test—our method still outperforms textual space synthesis by a significant margin, achieving an absolute reward gain of 0.4. This non-trivial improvement indicates that our method has not yet saturated at this scale. We will amend this result in our manuscript - thanks again for the helpful comment!
>
>
> | Sample numbers | Textual space synthesis | Latent-space synthesis |
> | -------- | -------- | -------- |
> | 50K     | 1.83     | **2.31** ($\uparrow0.4$)    |
> | 100K     |  2.01    | **2.41** ($\uparrow0.4$)    |

---

> > ### Author Response · Authors · 2025-08-06
> >
> > Dear reviewer 8VC1,
> >
> > Thank you once again for taking the time to review our paper.
> >
> > Following your feedback, we have worked diligently to address your  primary concerns, and in particular, present more detailed experimental setup as well as the additional empirical results.
> >
> > With the discussion deadline approaching, please let us know if you have any further questions — we'd be happy to respond. We are also committed to revising the paper based on your suggestions to improve its quality.
> >
> > Thank you for your time and consideration.
> >
> > Best,
> > Authors.

---

> > ### Comment · Reviewer_8VC1 · 2025-08-07
> > **Thanks for the rebuttal and sorry for the delay**
> >
> > I am very happy with the rebuttal and would like to thank the authors for running the additional experiments.
> > Please include them in the final version of the paper.
> > I will hence raise my score.
> > The main reason i am not raising the score further is because i still believe that a concrete comparison with active learning methods is crucial for any practical use. In a real setting, one should always be able to access some sort of new data annotation if not for user drift alone already.
> > However i think the above has well addressed my concerns

---

> > > ### Author Response · Authors · 2025-08-07
> > >
> > > Thank you so much for taking the time to read our rebuttal. We are glad that our response addresses your concerns and we appreciate your insightful comments and support! We would include them in the final version of the paper.
> > >
> > > Best, Authors

---

### Official Review · Reviewer_7482 · 2025-07-01

**Clarity:** 3
**Significance:** 3
**Originality:** 3
**Rating:** 4
**Confidence:** 4

**Summary:**

The paper proposes an approach to perform data augment for reward modeling. While existing approaches augment dataset of reward modeling by using LMs to generate new responses as text, the authors proposed to generate feature embeddings. This is less computationally expensive than generating text.

**Questions:**

Is there a way to perform quality control on the generated embeddings? For example, can we approximately reverse them and trace them back to similar textual-space sequences in a database?

**Ethical Concerns:**

["NO or VERY MINOR ethics concerns only"]

**Final Justification:**

After reading and replying to the authors' rebuttal, I'd like to maintain my assessment.

**Limitations:**

See "Strengths And Weaknesses"

**Quality:**

3

**Strengths And Weaknesses:**

## Strength
The paper is well written. The limitation of textual space synthesis is well explained and motivated. The proposed approach comes with theoretical guarantees.

## Weakness
The proposed approach operates purely in the embedding space, which is efficient. But in this case, the augmented positive and negative samples are not directly interpretable as text, and thus lack interpretability. When using a reward model for safety-critical alignment, one would want to know what constitutes positive and negative training examples. But the latent-space synthesis approach doesn't provide such a text-based interpretation of data. This lack of interpretability also makes it hard to evaluate the quality of the generated embeddings, e.g., whether they are sufficiently diverse or contain unwanted attributes.

Another limitation of the proposed approach is that it is based on a model trained on static embeddings. If one wants to try other prompts or incorporate other preference data, for example, this is simple with text-space synthesis. For latent-space synthesis, this requires retraining the VAE.

---

> ### Author Rebuttal · Authors · 2025-07-29
>
> We thank the reviewer for the constructive and helpful feedback, and for recognizing the value of our work. Below we address your concerns in detail.
>
> > W1/Q1: lack of interpretability and quality control
>
> You are raising a very thoughtful comment here. We fully agree that understanding and evaluating the quality of positive and negative examples is essential, and interpretability provides a valuable lens for this.
>
>
> That said, we believe interpretability—while important—is not the only route to ensuring quality. Our method takes a more principled and rigorous approach by providing theoretical guarantees on the quality of synthesized preference pairs. Specifically, our main result (Theorem 1) shows that under mild regularity assumptions, latent-space perturbations preserve preference ordering with high probability, and the resulting synthesized pairs maintain semantic consistency with the original labeled data. **This provides a provable basis for quality assurance that complements interpretability-driven techniques**.
>
> Moreover, we empirically validate the quality of the generated embeddings along multiple axes—e.g., improved reward model performance (Table 1), generalization across different model families and tasks (Table 1 and 4), and latent space quality via t-SNE visualization (Fig. 3). These evaluations offer evidence that our synthesized embeddings are meaningful, even without decoding them into text.
>
> We do acknowledge that interpretability of latent embeddings remains an open challenge, and we are excited about possible extensions. For instance, techniques such as [1] may help project synthesized embeddings back into text space for inspection. We view our method as complementary to such interpretability tools, and incorporating them is a promising direction for future work.
>
>
> [1] Belrose et al., Eliciting Latent Predictions from Transformers with the Tuned Lens, arXiv preprint 2303.08112
>
>
>
> > W2: The need for retraining the VAE with other prompts.
>
> We agree with the reviewer that our latent-space synthesis method is based on a model trained on static embeddings, and adapting to substantially different prompts or new preference data may require retraining the VAE. This is a valid limitation, and we have **explicitly acknowledged it in our Limitations section (Appendix C) with full transparency**.
>
> That said, we would like to emphasize that **retraining the VAE is computationally very lightweight and fast**—our model has only 0.5M parameters and can be trained efficiently even on modest hardware. In contrast, adapting text-space synthesis methods typically requires rerunning large-scale generation and annotation with LLMs, which is far more resource-intensive.
>
> We will clarify this trade-off more explicitly in the final version and appreciate the reviewer’s thoughtful feedback.

---

> ### Comment · Reviewer_7482 · 2025-08-04
> **Reply to the rebuttal**
>
> Thanks for your feedback. I agree with your response to the interpretability problem that I mentioned:
> > That said, we believe interpretability—while important—is not the only route to ensuring quality
>
> I'd like to maintain my score (borderline accept).

---

> > ### Author Response · Authors · 2025-08-04
> >
> > Thank you so much for taking the time to read our rebuttal. We are glad that our response addresses your concerns and we appreciate your insightful comments and support!
> >
> > Best,
> > Authors

---

### Official Review · Reviewer_5d5f · 2025-07-02

**Clarity:** 4
**Significance:** 3
**Originality:** 4
**Rating:** 5
**Confidence:** 4

**Summary:**

This paper tackles data scarcity in training reward models for aligning LLMs with human preferences. Rather than generating additional text, and then labelling by humans/powerful LLMs, the authors train a variational autoencoder to learn a structured latent space and then sample diverse synthetic preference pairs directly from that space. This approach yields faster, more cost-effective data generation. Experiments from multiple perspectives show that the method delivers substantial gains over baseline techniques.

**Questions:**

*  In Section 3.1, how does the choice of embedding layer influence generation performance?

* Could you provide additional details on how the top-k synthetic latent codes are selected, including the computation of their likelihood scores?

* Table 1 shows continued performance gains as the augmentation scale increases. Could you test higher augmentation rates to determine where the performance begins to plateau?

* Table 4 indicates that the gap between the original and augmented data narrows as model size grows. Up to what model size does your method remain beneficial?

**Ethical Concerns:**

["NO or VERY MINOR ethics concerns only"]

**Final Justification:**

The paper is well-written and self-contained. The idea of synthesizing data for efficient alignment is novel. While human feedback is often treated as gold data, it can be very diverse and hard to learn. Instead, we can learn a certain distribution from data and then sample synthetic representations from the latent space.

**Limitations:**

yes

**Paper Formatting Concerns:**

N.A.

**Quality:**

4

**Strengths And Weaknesses:**

Strengths:

* The paper tackles the urgent problem of data scarcity in preference alignment with a clear and timely focus.

* By training a VAE and sampling synthetic preference pairs from its latent space, the authors achieve both efficiency and strong performance. These results are impressive and could reshape future data-generation practices.

* The approach is technically solid and supported by  theoretical proof and extensive experiments.

* The manuscript is well written, clearly organized, and easy to follow.

Weaknesses:
This is a good paper; however, the following questions require clarification.

---

> ### Author Rebuttal · Authors · 2025-07-29
>
> We thank the reviewer for the constructive and helpful feedback, and for recognizing the value of our work. Below we address your concerns in detail.
>
> > Q1: The choice of embedding layer
>
> This is a great question! We use the last-layer embeddings for synthesis because the reward model predictions are directly based on this layer. To verify this choice, we conducted ablation studies on different embedding layers, and the last-layer embeddings consistently yielded the best generation performance. We report the results based on Llama-3-8B on HH-RLHF with 4x augmentation.
>
>
>
> | Layer | Performance |
> | -------- | -------- |
> | 1      |   0.22   |
> | 10 (middle)     |    1.02  |
> | 20 (middle)     |   1.37   |
> | 32 (last)     |  1.94    |
>
>
> > Q2: Additional details on how the top-k synthetic latent codes are selected
>
> The selection of the top-k synthetic latent embedding works as follows:
>
> 1. **Generate candidates**: For each original latent vector (representing a preferred or non-preferred response), several candidate latent codes are created by adding Gaussian noise.
> 2. **Compute likelihood scores**: Each candidate latent code is evaluated based on how likely it is under the VAE’s learned Gaussian distribution. This is done by checking how close the candidate is to the mean of the distribution, with smaller deviations resulting in higher likelihood scores.
> 3. **Select top-k**: From all the candidates, the $k$ codes with the highest likelihood scores are chosen. This ensures that only the most plausible synthetic latent codes (those that are close to the high-density regions of the learned latent space) are kept and later decoded into synthetic embeddings.
>
> We will revise our manuscript to make this process clearer - thanks for pointing this out!
>
> > Q3: At what point does the performance begin to plateau for the augmentation scale increases?
>
> We have conducted ablation studies on varying augmentation scales and observed that performance improvements gradually taper off beyond the tested augmentation rates.  We report the results based on Llama-3-8B on HH-RLHF with different augmentation rates. Additional experiments with even higher augmentation scales, as large as 32x, are the point where performance saturates.
>
>
>
> | Augmentation scale | Gold reward |
> | -------- | -------- |
> | 2X     |   1.86   |
> | 4X     |   1.94   |
> | 8X     |   2.20   |
> | 16X     |    2.28  |
> | 32X     |   **2.32**   |
> | 64X     |   2.26   |
>
>
> > Q4: At what model size does your method remain beneficial?
>
> We extended our experiments to larger models, including Qwen/Qwen2.5-14B. Our method continues to provide significant benefits at this scale. Due to computational constraints, we were unable to conduct experiments on models larger than 70B. We report the results based on Llama-3-8B on HH-RLHF with 4x augmentation.
>
>
>
> | Qwen2.5-14B | Gold reward |
> | -------- | -------- |
> | Textual space synthesis     | 1.74     |
> | Latent space synthesis     | **2.05**     |

---

> > ### Comment · Reviewer_5d5f · 2025-08-08
> >
> > Thanks for the clarification. I maintain my score.

---

> > > ### Author Response · Authors · 2025-08-08
> > > **Official Comment by Authors**
> > >
> > > Thank you so much for taking the time to read our rebuttal and maintaining your score. We are glad that our response clarifies your concerns and we appreciate your insightful comments and support!
> > >
> > > Best,
> > >
> > > Authors

---

### Decision · Program_Chairs · 2025-09-17

**Decision:**

Accept (poster)

**Comment:**

The paper presents a fast, resource-efficient method for generating preference data for reward modelling. The approach uses a Variational Autoencoder in the model’s latent space, producing diverse, high-quality pairs and improving model performance compared to previous methods.

Strength of the work: The approach achieves strong performance and computational gains over other methods, is well supported by extensive experiments, and the paper is clearly written and well organized.

The detailed rebuttal effectively addressed the reviewers' concerns. Therefore, the final decision is to accept the paper.